# USP26 functions as a negative regulator of cellular reprogramming by stabilising PRC1 complex components

Bo Ning[1], Wei Zhao[1,2], Chen Qian[1], Pinghua Liu[1], Qingtian Li[1], Wenyuan Li[1,3] & Rong-Fu Wang[1,4]

Despite much progress in the comprehension of the complex process of somatic cell reprogramming, many questions regarding the molecular mechanism of regulation remain to be answered. At present, the knowledge on the negative regulation of reprogramming process is indeed poor in contrary to the identification of positive regulators. Here we report for the first time that ubiquitin-specific protease 26 negatively regulates somatic cell-reprogramming process by stabilizing chromobox (CBX)-containing proteins CBX4 and CBX6 of polycomb-repressive complex 1 through the removal of K48-linked polyubiquitination. Thus, accumulated CBX4 and CBX6 repress the expression of pluripotency genes, such as *Sox2* and *Nanog*, through PRC1 complexes to ubiquitinate histone H2A at their promoters. In all, our findings have revealed an essential role for ubiquitin-specific protease 26 in cellular reprogramming through polycomb-repressive complex 1.

[1] Center for Inflammation and Epigenetics, Houston Methodist Research Institute, Houston, TX 77030, USA. [2] Key Laboratory for Stem Cells and Tissue Engineering, Ministry of Education, Sun Yat-sen University, Guangzhou 510080, China. [3] Xiangya Hospital, Xiangya School of Medicine, Central South University, Changsha 410008, China. [4] Department of Microbiology and Immunology, Weill Cornell Medical College, Cornell University, New York, NY 10065, USA. Correspondence and requests for materials should be addressed to R.-F.W. (email: rwang3@houstonmethodist.org)

Somatic cells can be reprogrammed by the transduction of four key transcription factors (*Oct4*, *Sox2*, *Klf4*, and *cMyc*) to give rise to induced pluripotent stem cells (iPSCs)[1–3]. With a view to utilizing iPSCs for the studies involving regenerative medicine and drug screening, many researchers have focused on the dissection of the molecular mechanisms of cellular reprogramming[4]. However, this reprogramming process is inefficient and variable. Therefore, understanding other factors particularly negative regulators of cellular reprogramming should be instrumental for developing a more reliable and accelerated process for obtaining iPSCs. In the last few years, many epigenetic factors have been identified to play critical roles and reprogramming somatic cells into a pluripotent state[5–7]. For instance, histone methyltransferase and demethylase, such as Wrd5, SUV39H1/2, Setdb1, Utx, Jmjd3, and Dot1L, either positively or negatively regulate the kinetics and efficiency of cellular reprogramming[6–10]. Despite theoretical role of PRC1 in the repression of specific genes in differentiation[11], the precise mechanisms that control the dynamic of the various protein subunits of the PRC1 complexes during cellular reprogramming are poorly understood.

The core proteins of PRC1 complex comprises RING1A or RING1B with one of six polycomb group RING finger (PCGF) proteins, which can bind to RING1A or RING1B within the E3 catalytic unit of the PRC1 complex[12]. On the basis of the composition of various protein subunits, these PRC1 complexes are classified as PRC1.1–PRC1.6 families[13]. PRC1 complexes can ubiquitinate histone 2A lysine 119 (H2AK119), repressing cell lineage-specific or pluripotency gene transcription[14, 15]. The canonical PRC1 variants (PRC1.2 and PRC1.4) contain PCGF2 and PCGF4, respectively, RING1A/B, polyhomeotic (PHC) and chromobox-containing (CBX) proteins, such as CBX2, CBX4, CBX6, or CBX7, and can specifically recognize H3K27me3 on H3 histone[12]. The dynamic interchange of PRC1.2 and PRC1.4 subunits modulates the balance between self-renewal and lineage commitment in embryonic stem cells (ESCs)[16].

In this study, we report that the post-translational regulation of PRC1 components CBX4 and CBX6 by ubiquitination is critical for reprogramming. Importantly, our systematic investigation demonstrates that deubiquitinase ubiquitin-specific protease 26 (USP26) acts as a potent negative regulator in the process of somatic cell reprogramming into iPSCs by stabilizing CBX4 and CBX6 through the USP26-mediated removal of K48-linked ubiquitination. We further show that the ectopic expression of *Usp26* blocks reprogramming by repression of pluripotency genes, such as *Sox2* and *Nanog*, mediated through CBX4 and CBX6 accumulation. By contrast, knockdown of *Usp26* enhances the efficiency of reprogramming by reactivating *Sox2* and *Nanog* through degradation of CBX4 and CBX6. To our knowledge, this is the first demonstration of the coordination of the USP26-CBX4/CBX6 axis in the negative regulation of somatic cell reprogramming, thus providing new insights into molecular mechanisms by which USP26 regulates the specific components of PRC1 complexes.

## Results

**Functional role of USP26 in cellular reprogramming.** Recent studies show that the ubiquitin (Ub)–proteasome system regulates stem cell pluripotency and cellular reprogramming by ubiquitination-mediated degradation of key pluripotency factors[17], but the function and mechanisms of many USP proteins in cellular reprogramming remain poorly understood. To define the potential functions of the USP family proteins in cellular reprogramming, we screened USP family members, using short hairpin RNA (shRNA) knockdown, based on their ability to

increase or inhibit the reprogramming efficiency of mouse embryonic fibroblast (MEF) into iPSCs. For this purpose, shRNAs for 48 mouse *Usp* family members or reference shRNA

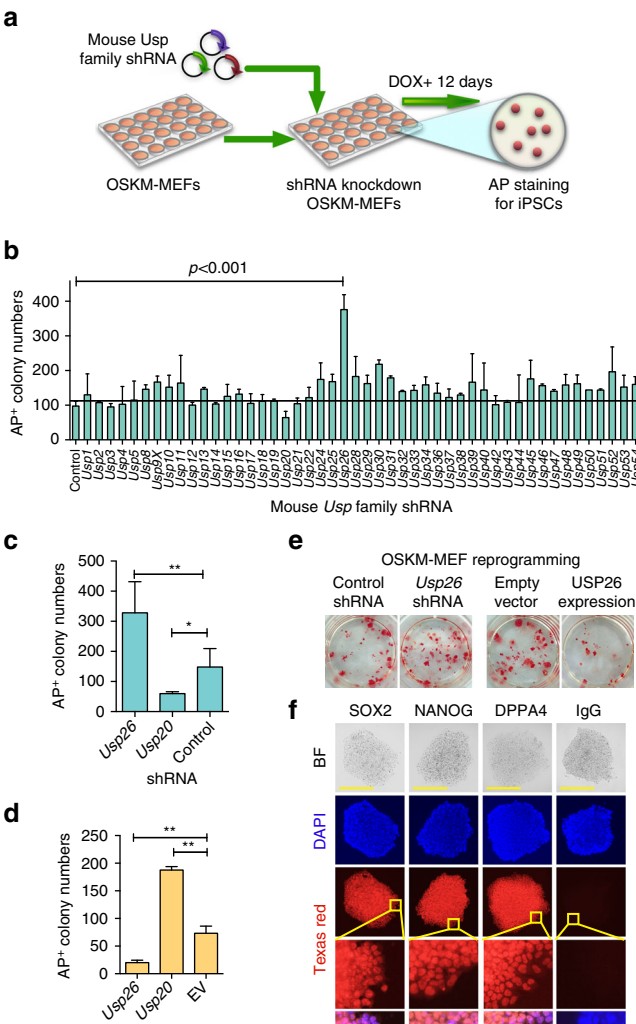

**Fig. 1** Identification of USP26 as a negative regulator of pluripotency. **a** Schematic of experimental strategy for screening essential *Usps* for the generation of iPSCs. Dox-inducible OSKM-transgenic MEFs were plated, transfected with 48 individual mouse *Usp* shRNA lentiviral vectors to knockdown *Usps*, and stained for AP+ iPSC colonies after 12 days of Dox treatment. **b** Quantification of AP+ colonies after 12 days of OSKM induction in MEFs transduced with shRNA, targeting members of the mouse *Usp* family, $p < 0.001$ compared to control shRNA. **c** Quantification of AP+ colonies after 12 days of OSKM induction in MEFs transduced with mouse *Usp26* shRNA, *Usp20* shRNA, or control shRNA lentivirus, *$p < 0.05$ compared to control shRNA, **$p < 0.01$ compared to control shRNA. **d** Quantification of AP+ colonies after 12 days of OSKM induction in MEFs transduced with pLtet-O (tetracycline-inducible) mouse *Usp26*, *Usp20*, or empty vector (EV) lentivirus, **$p < 0.01$ compared to EV. **e** Images of AP staining of iPSC colonies after 12 days' OSKM induction in MEFs transduced with control shRNA, *Usp26* shRNA, EV, or pLtet-O lentivirus overexpressing *Usp26*. **f** Bright field (BF) and immunofluorescence microscopic images of Oct4, Nanog, and Dppa4 in *Usp26* knockdown iPSCs. iPSC colonies were fixed, blocked, and stained with specific antibodies, followed by goat anti-mouse antibody-conjugated Texas Red. Nuclei were stained with DAPI. *Scale bar*, 100 μM. The data are presented as means ± SD from three independent experiments. **b–d** Two-way ANOVA for multiple comparisons

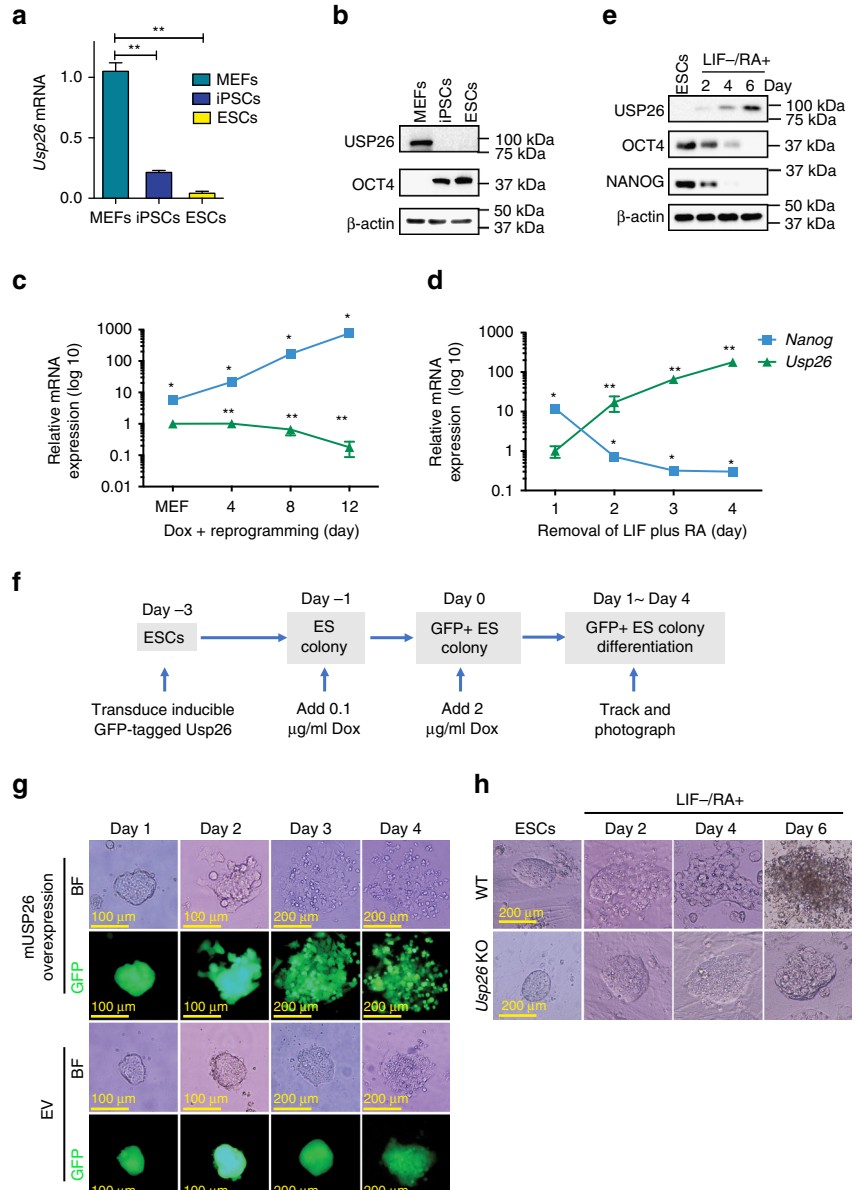

**Fig. 2** USP26 is required for ESC differentiation. **a** Real-time qPCR analysis of mouse *Usp26* mRNA expression in MEFs, iPSCs, and ESCs, **\*\****p* < 0.01 compared to MEFs. **b** Western blot analysis of USP26 protein expression in MEFs, iPSCs, and ESCs. **c** Real-time qPCR analysis of mouse *Usp26* and *Nanog* mRNA expression during Dox-induced OSKM-mediated MEF reprogramming on days 0, 4, 8, and 12, *\*p* < 0.05 compared to *Usp26*, **\*\****p* < 0.05 compared to day 0. **d** Real-time qPCR analysis of mouse *Usp26* and *Nanog* mRNA expression in ESCs after LIF withdrawal and treatment with 1 μM RA on days 1, 2, 3, and 4, *\*p* < 0.05 compared to *Usp26*, **\*\****p* < 0.05 compared to day 1. **e** Western blot analysis of USP26, OCT4, and NANOG protein expression in ESCs after LIF withdrawal and treatment with 1 μM RA at days 0, 2, 4, and 6. β-actin was used as a loading control. **f** Experimental scheme of Usp26-induced ESC differentiation. ES cells were transduced with Dox-inducible GFP-tagged Usp26 or empty vector lentivirus. After ESC colonies formed, then low-dose Dox (0.1 μg/ml) was added for 1 day pre-selection. Day 0 was defined as the day when pre-selected GFP-positive colonies were cultured in iSF1 medium with high-dose Dox (2 μg/ml), individual colonies were tracked and taken pictures on Usp26-expressing days 1, 2, 3, and 4. **g** Bright field (BF) and fluorescent microscopic images of ESC morphology with Dox-inducible GFP-tagged m*Usp26* overexpression or with GFP-tagged empty vector (EV). **h** Bright field images of ESC morphology of wild-type (WT) or *Usp26* knockout (KO) ESCs treated with RA. WT or *Usp26* KO ESCs were cultured in ES differentiation medium (LIF withdrawal with 1 μM RA). Individual colonies were tracked and photographed on days 0, 2, 4, and 6 under microscope. The data are presented as means ± SD from three independent experiments. **a**, **c**, **d** Two-way ANOVA for multiple comparisons

from Dharmacon mouse shRNA library (with a GFP expression cassette in the vector) were selected based on their ability to knockdown their corresponding genes. We used at least two individual shRNAs of each gene with high knockdown efficiency (>70%) for screening. Dox-inducible Oct4-Sox2-Klf4-cMyc (OSKM) transgenic MEFs were transduced with individual lentiviral shRNAs specific for USP gene. The reprogramming efficiency was determined by alkaline phosphatase (AP) staining

after doxycycline (Dox) induction for 12 days (Fig. 1a). AP staining showed that knockdown of *Usp26* in MEFs (Supplementary Fig. 1a) generated ~400 iPSC colonies, which was approximately fourfold higher than MEFs transduced with control shRNA (Fig. 1b, c). Conversely, iPSC colony numbers decreased upon *Usp20* knockdown (Fig. 1c). The knockdown of other Usp family member proteins, such as USP30 and USP52, was also found to exert effects on reprogramming efficiency,

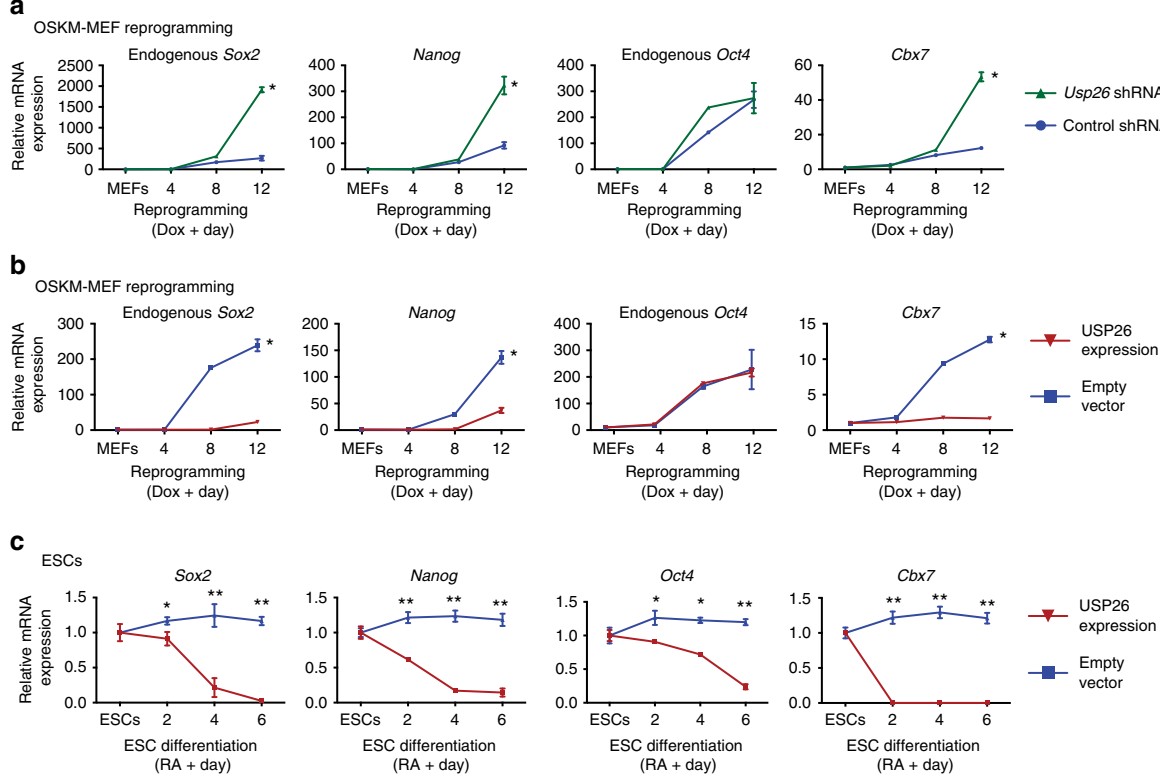

**Fig. 3** USP26 inhibits pluripotency and polycomb gene expression. **a** Real-time PCR (qPCR) analysis of mouse *Sox2*, *Nanog*, *Oct4*, and *Cbx7* mRNA expression in *Usp26* or control shRNA lentivirus-transduced OSKM MEFs (*$p < 0.05$ compared to control shRNA). **b**, **c** Real-time PCR (qPCR) analysis of mouse *Sox2*, *Nanog*, *Oct4*, and *Cbx7* mRNA expression in OSKM MEFs **b** and ESCs **c** transfected with mouse *Usp26* or empty vector (EV; *$p < 0.05$ compared to EV, **$p < 0.01$ compared to EV). The data are presented as means ± SD from three independent experiments. **a**–**c** Two-way ANOVA for multiple comparisons

albeit at much lesser extent (less than twofold). In contrast, the overexpression of *Usp26* decreased the number of AP⁺ colonies (Fig. 1d, e), while overexpression of *Usp20* increased the number of AP⁺ colonies (Fig. 1d). The previous study has shown that *Usp20* increases the expression of HIF1A[18, 19], which is reported as a metabolic switch for an early stage of iPSC[20]. *Usp26* shRNA transduction efficiency was characterized by shRNA GFP expression in iPSC colonies (Supplementary Fig. 1d) and by quantitative polymerase chain reaction (qPCR) of *Usp26* (Supplementary Fig. 1a). The reprogramming efficiency of USP26 was also evaluated by NANOG immunofluorescent staining (Supplementary Fig. 1b, c). The role of USP26 in somatic cell reprogramming was further confirmed by the development of teratomas, containing several tissues, such as adipose tissue, neural rosette-like tissue, and gut-like epithelial tissue (Supplementary Fig. 1e) and by protein immunostaining of pluripotency markers, such as OCT4, NANOG, and DPPA4 (Fig. 1f). Cumulatively, our data suggest that USP26 plays an important role in cellular reprogramming into a pluripotent state.

**USP26 expression promotes ESC differentiation**. To study the function of USP26 in pluripotency, we compared mRNA and protein levels of USP26 in MEFs, ESCs, and iPSCs. *Usp26* mRNA and protein levels in MEFs were higher than those in iPSCs or ESCs (Fig. 2a, b). As expected, *Nanog*, a known pluripotency gene, increased dramatically during Dox-induced OSKM MEF reprogramming and decreased upon removal of leukemia inhibitory factor (LIF) during retinoic acid (RA)-induced ESC differentiation (Fig. 2c, d). However, *Usp26* mRNA gradually decreased during Dox-induced OSKM-mediated MEF reprogramming and increased during ESC differentiation (Fig. 2c, d).

Similarly, USP26 protein levels increased from day 0 to day 6 upon addition of RA (Fig. 2e). Furthermore, infection of ESCs with GFP-tagged mouse (m) USP26 led to ESC differentiation in an RA-independent manner (Fig. 2f). To further understand the role of USP26 in ES cells, we generated *Usp26* knockout ESCs using a CRISPR/Cas9 technology. Since the expression level of USP26 is low in ESCs, we tested the knockout efficiency of *Usp26* using MEFs (Supplementary Fig. 2c, d). Wild-type (WT) ESCs began differentiating within 2 days after RA stimulation, whereas *Usp26* knockout ESCs maintained ESC morphology even after incubation with RA for 6 days (Fig. 2g) and began differentiating on day 8 (Supplementary Fig. 2a). However, protein levels of pluripotency and Polycomb markers, such as NANOG and CBX7, in *Usp26* knockout cells, remain similar during a 6-day RA induction of ESCs (Supplementary Fig. 2b). These results suggest that USP26 expression promotes ESC differentiation.

**USP26 inhibits expression of pluripotency core genes**. To study the effects of *Usp26* knockdown on gene expression during MEF reprogramming, we performed qPCR and measured the expression levels of 90 genes, which have been reported to be involved in reprogramming (Supplementary Data 1). Our results showed that after *Usp26* knockdown during Dox-induced OSKM-mediated MEF reprogramming led to increased mRNA levels of *Sox2* and *Nanog*, but not *Oct4*, compared with control (Fig. 3a). Overexpression of *Usp26* inhibited *Sox2* and *Nanog* gene expression during Dox-induced OSKM-mediated MEF reprogramming (Fig. 3b). These data suggest that the expression of *Sox2* and *Nanog* gene is regulated by *Usp26* during Dox-induced OSKM-mediated MEF reprogramming. Overexpression of *Usp26* in ESCs led to decreased mRNA levels of

*Sox2*, *Nanog*, and *Oct4* compared with control cells (Fig. 3c). This inhibition of pluripotent gene expression began as early as 2 days after induction of ESC differentiation without RA. These results suggest that USP26 regulates the expression of pluripotency genes during cellular reprogramming to a pluripotent state and ESC differentiation. We also found significant decrease in the gene expression of *Cbx7*, a Polycomb protein, upon *Usp26*-induced ESC differentiation (Fig. 3c). Upon Dox-induced OSKM-mediated MEF reprogramming, *Cbx7* gene expression increased when *Usp26* was knocked down (Fig. 3a) and decreased when *Usp26* was overexpressed (Fig. 3b). In addition, based on qPCR analysis (Supplementary Table 1), *Cbx7*, *Nanog*, and *Sox2* were identified as the most significantly changed genes in MEFs during reprogramming with *Usp26* knockdown. Overall, these data suggest that the PRC1 components are potential targets of USP26.

**USP26 binds to CBX4 and CBX6 in RING1A-independent manner.** To determine whether USP26 targets subunits of the PRC1 complexes, we performed immunoprecipitation (IP) experiments to screen for protein–protein interactions. Hemagglutinin (HA)-tagged PRC1 components, including RING1A, RING1B, RYBP, PCGF1, PCGF2, BM1, CBX4, CBX6, CBX7, and CBX2, were transfected into 293T cells along with FLAG-tagged human (h) USP26. Components of the PRC1.2 complex including RING1A, PCGF2, CBX4, CBX6, and CBX7 interacted with USP26, but components of other PRC1 families, such as RING1B (all PRC1s), RYBP (all PRC1s), PCGF1 (PRC1.1), BMI1 (PRC1.4; Fig. 4a) as well as KDM2B (PRC1.1) and PHC1 (PRC1.4; Supplementary Fig. 3a), did not interact with USP26. Components of the PRC2 complex EZH1, EED, and EZH2 were also tested, but no interactions were observed (Supplementary Fig. 3a). Since the undifferentiated ESCs expressed CBX7 but no USP26, endogenous interactions between USP26 and CBX4 or CBX6, but not CBX7, were only observed in differentiating ESCs (Fig. 4b). To determine whether USP26 interacts with the PRC1 components RING1A, CBX4, and CBX6 independent of each other, we generated RING1A, CBX4, or CBX6 knockout 293T cells using CRISPR/Cas9 technology (Supplementary Fig. 3c, d). IP experiments showed binding of FLAG-USP26 and CBX4 or CBX6 in RING1A KO 293T cells (Fig. 4c), suggesting that USP26 and CBX4 or CBX6 interactions are RING1A-independent. However, in CBX4 or CBX6 knockout 293T cells, USP26 did not interact with RING1A, suggesting that USP26 and RING1A interactions are dependent on the presence of CBX4 or CBX6 (Fig. 4c). These protein–protein interactions in RING1A KO, CBX4 KO, or CBX6 KO 293T cells were further demonstrated using HA-tagged PRC1 proteins and FLAG-tagged USP26 (Supplementary Fig. 3b, e, f). Taken together, these results suggest that USP26 physically binds to components of the PRC1 complex, including RING1A, CBX4, and CBX6, and the interaction between USP26 and CBX4 or CBX6 is independent of the presence of RING1A.

**USP26 stabilizes CBX4 and CBX6.** To further identify molecular mechanisms of how USP26 regulates CBX4 and CBX6, we measured *Cbx4* and *Cbx6* mRNA and protein levels in MEFs with or without *Usp26* overexpression. Although *Usp26* ectopic expression in MEFs did not change *Cbx4* or *Cbx6* mRNA levels (Fig. 5a and Supplementary Fig. 4a), we found that *Usp26* expression resulted in increased CBX4 and CBX6 protein levels (Fig. 5b and Supplementary Fig. 4b), thus suggesting that *Usp26* regulates CBX4 and CBX6 in a post-transcriptional manner. Furthermore, with increasing expression of Dox-inducible USP26 in 293T cells (Supplementary Fig. 4c) and in ES cells (Fig. 5c and Supplementary Fig. 4d), protein levels of CBX4 and CBX6 were

also increased. Additionally, CBX4 and CBX6 protein levels were also significantly increased upon addition of the proteasome inhibitors, MG132 or Lactacystin in 293T cells (Supplementary Fig. 4e). In MEF cells, which express USP26 endogenously, we also found that proteasome inhibitors have less effect on CBX4 and CBX6 accumulation (Supplementary Fig. 4f), USP26 stabilize CBX4 and CBX6 by reducing proteasomal degradations. Protein modifications by K48-linked poly-Ub chains are well-established signals for recognition and initiation of degradation by the 26S proteasome complexes. Therefore, we analyzed K6-linked, K11-linked, K27-linked, K29-linked, K33-linked, K48-linked, and K63-linked polyubiquitination of CBX4 and CBX6 by IP (Supplementary Fig. 4g). We found that both CBX4 and CBX6 were modified by K48-linked polyubiquitination, which could be removed by USP26 (Fig. 5d). We also generated a catalytically inactive USP26 mutant by replacing the active-site cysteine with a serine residue (C304S), and a deletion mutant by deleting the conserved domain (Δ295-312). Neither of these USP26 mutants could remove K48-linked polyubiquitination of CBX4 or CBX6 (Fig. 5e), indicating that the deubiquitinase activity of USP26 is required for the regulation of CBX4 or CBX6 protein stability.

**Accumulated CBX4 and CBX6 inhibit pluripotent genes.** Next, we sought to determine whether increased CBX4 and CBX6 levels affect pluripotent genes. ChIP–qPCR of the promoters of *Sox2*, *Nanog*, *Oct4*, and *Cbx7* in *Usp26*-differentiated ESCs on day 6 revealed binding of USP26, CBX4, CBX6, H2A-ubi1, and H3K27me3 to the *Sox2*, *Nanog*, and *Cbx7* promoters (Fig. 6a), suggesting that promoter-binding of these components led to a decrease in pluripotency gene expression (Fig. 3c). Similarly, during RA-induced ESC differentiation, USP26, CBX4, CBX6, H2A-ubi1, and H3K27me3 also bound to the *Sox2*, *Nanog*, and *Cbx7* promoters (Supplementary Fig. 5a). However, only H3K27me3 occupied the promoter of *Oct4* (Fig. 6a and Supplementary Fig. 5a). In addition, the core components of PRC1.2, RING1A, and PCGF2 were also localized to *Sox2*, *Nanog*, and *Cbx7* promoters on day 0 and day 6 of ESC differentiation (Supplementary Fig. 5a). This suggests that RING1A and PCGF2 heterodimers form stable complexes at the *Sox2*, *Nanog*, and *Cbx7* promoters during pluripotency maintenance and differentiation. Upon ESC differentiation, other components, such as USP26, CBX4, and CBX6, may bind to these core RING1A and PCGF2 heterodimers to constitute the PRC1.2 complex, which can specifically recognize H2A-ubi1 and H3K27me3 to repress *Sox2*, *Nanog*, and *Cbx7* gene transcription[12, 21]. Conversely, during *Usp26*-induced ESC differentiation and RA-induced ESC differentiation, the *Cbx4* and *Cbx6* promoters bound less H2A-ubi1 and H3K27me3, which was associated with less CBX7 binding (Fig. 6b) and less RING1A and PCGF2 binding (Supplementary Fig. 5a). Increased *Cbx4* and *Cbx6* mRNAs were also observed in *Usp26*-induced or RA-induced ESC differentiation (Supplementary Fig. 5b). CBX4 was also previously reported to silence *Cbx7*, which, in turn, may relieve *Cbx4* and *Cbx6* gene repression[22], suggesting that CBX4 and CBX6 may form an amplification loop to permit their continuous expression during ESC differentiation. This result suggests that the increased *Usp26* expression during ESC differentiation may lead to increased CBX4 and CBX6 protein expression not only by CBX4 and CBX6 protein stabilization but also indirectly by reversing the repression of their gene expression. We further confirmed that CBX4 and CBX6 repressed the promoter activities of *Sox2*, *Nanog*, and *Cbx7* using a luciferase assay (Fig. 6c). Furthermore, we analyzed *Sox2*, *Nanog*, and *Cbx7* promoters in *Cbx4* and/or *Cbx6* knockdown ES cells. ChIP–qPCR analyses revealed the recruitment of Usp26 and H2A-ubi1 to these promoters (Supplementary Fig. 5e), suggesting the repression of *Sox2*, *Nanog*, and *Cbx7*

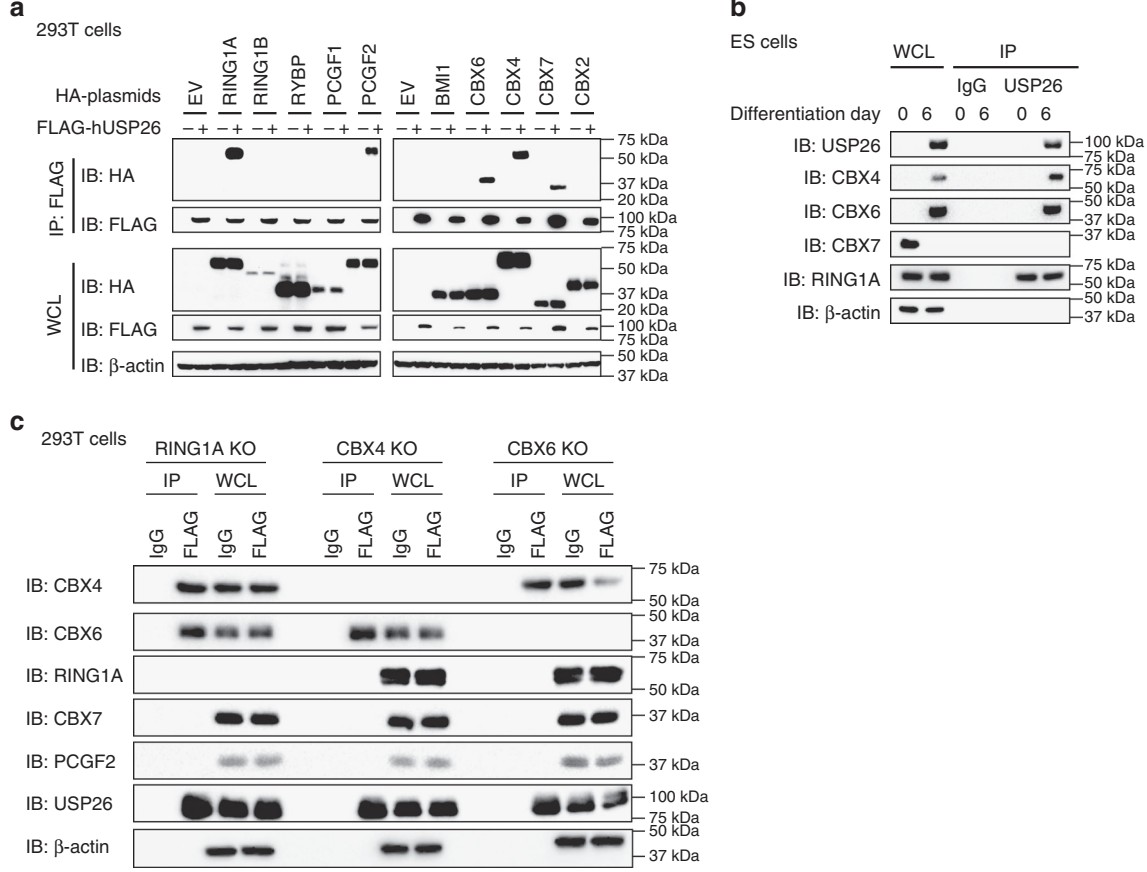

**Fig. 4** USP26 interacts with PRC1 components. **a** 293T cells were transfected with HA-tagged RING1A, RING1B, RYBP, PCGF1, PCGF2, BMI1, CBX4, CBX6, CBX7, or CBX2 plus FLAG-hUSP26 or empty vector. Following IP with anti-FLAG beads, specific proteins were analyzed by western blotting using anti-HA antibody. **b** Mouse ESCs were infected with or without mouse USP26 lentivirus, and cell extracts were harvested at different time points, immunoprecipitated with anti-USP26 antibody, and analyzed by western blotting using specific antibodies, as indicated. **c** RING1A, CBX4, and CBX6 knockout (KO) 293T cells were transfected with FLAG-USP26, cell extracts were harvested, immunoprecipitated with anti-FLAG antibody, and analyzed by western blotting using specific antibodies, as indicated

expression by Usp26. These results suggest that *Cbx4* or *Cbx6* alone is sufficient for downregulating *Sox2*, *Nanog*, or *Cbx7* expression. To understand how *Usp26* expression is regulated, we assessed the promoter-binding activity of *Usp26* by PRC1 subunits using the ChIP–qPCR assay during RA-induced ESC differentiation, and observed that PRC1 components PCGF2 and *Cbx7* occupation as well as H2A-ubi1 modifications decreased significantly in *Usp26* promoter (Supplementary Fig. 5d), thus indicating that CBX7 containing PRC1 complex is a potential repressor of *Usp26*. To study whether USP26 can regulate CBX7 directly or through interactions with CBX4 and CBX6, we performed luciferase assay and found that overexpression of USP26 alone could not inhibit promoter activities of these pluripotency genes. However, when USP26 was expressed with either Cbx4 or CBX6, significantly decreased promoter activities of *Sox2, Nanog*, and *Cbx7* were observed (Supplementary Fig. 5g), suggesting that USP26 interacts with *Cbx4* or *Cbx6* alone or both in *Cbx7* promoter for CBX7 suppression. Taken together, these results provide molecular mechanisms of Usp26-mediated increased CBX4 and CBX6 protein stability for suppressing pluripotent gene expression.

**Ectopic expression of CBX4 and CBX6 blocks reprogramming.** To determine how Polycomb proteins affect MEF reprogramming, we used shRNA to knockdown *Cbx4, Cbx6, Cbx7*, and *Ring1a* in MEFs. After 12 days of Dox treatment, knockdown of

*Cbx4* or *Cbx6*, but not *Cbx7* or *Ring1a*, led to an increased number of AP[+] colonies compared with the control shRNA group (Fig. 7a). These results, showing increased number of AP[+] colonies after *Cbx4* or *Cbx6* knockdown, were similar to the increased number of AP[+] colonies after *Usp26* knockdown (Fig. 1c), suggesting that Cbx4 or Cbx6 inhibits cellular reprogramming. *Sox2* and *Nanog* mRNA levels, but not *Oct4*, dramatically increased in *Cbx4* or *Cbx6* knockdown cells (Fig. 7b), similar to the increased *Sox2* and *Nanog* mRNA levels in *Usp26* knockdown cells (Fig. 3a). To determine the reprogramming efficiency of *Usp26* knockdown alone, and with different combinations of *Usp26* with *Cbx4* and/or *Cbx6* shRNAs, we performed similar experiments and found no significant differences in reprogramming efficiency between *Usp26* knockdown alone and different combined knockdown of *Usp26, Cbx4*, and *Cbx6*. Thus, our data indicate that CBX4 and *Cbx6* are downstream genes of *Usp26*. Furthermore, *Cbx4* or CBX6 alone could inhibit increased reprogramming efficiency mediated by Usp26 shRNA (Supplementary Fig. 6b). We also showed that *Cbx4* and *Cbx6* could promote ESC differentiation in *Usp26* knockout cells (Supplementary Fig. 6c), similar to results obtained by overexpressing *Usp26* (Fig. 2g). Overall, these results indicate that USP26 regulates CBX4 and CBX6 protein abundance through its deubiquitination activity during ESC differentiation, the increased amounts of CBX4 and CBX6 inhibits *Sox2* and *Nanog* expression, thus blocking cellular reprogramming and reducing pluripotency.

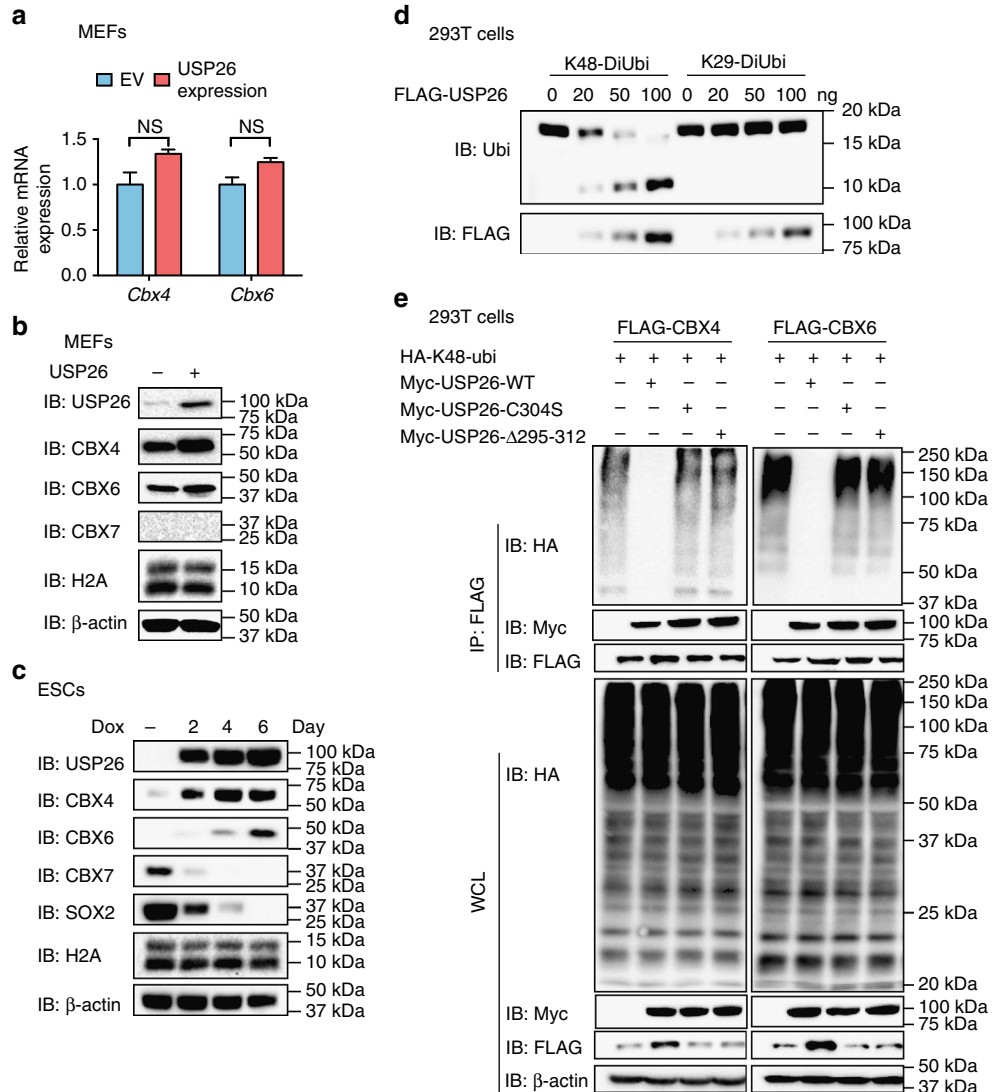

**Fig. 5** USP26 stabilizes CBX4 and CBX6 proteins by removing K48-linked polyubiquitination. **a**, **b** MEFs were transduced with mouse *Usp26* or EV lentivirus, cell extracts were harvested, and mRNA levels were analyzed by qPCR **a**. Protein levels were analyzed by western blotting with specific antibodies **b**, as indicated. **c** Mouse ESCs were infected with Dox-inducible m*Usp26* lentivirus, and cell lysates were harvested on days 0, 2, 4, and 6 and analyzed by western blotting using the indicated antibodies. **d** Deubiquitinase activity assay of USP26. Purified USP26 was incubated with K48-linked or K29-linked ubiquitination chains for 3 h. Cleaved Ub was detected by western blotting with anti-Ub antibody. **e** 293T cells were transfected with HA-tagged K48-linked Ub, FLAG-CBX4, or FLAG-CBX6 with or without Myc-tagged WT or Myc-tagged mutant (C304S or Δ295-312) USP26. Following IP with anti-FLAG beads, ubiquitination of CBX4 or CBX6 was analyzed by western blotting using anti-HA antibody. The data are presented as means ± SD from three independent experiments. **a** Unpaired two-tailed Student's *t*-test

## Discussion

The development of strategies to more efficiently reprogram somatic cells into a pluripotent state has broader applicability for disease modeling and tissue regeneration. The epigenetic modifier PRC1 is directly involved in regulating gene expression. The components of PRC1 complexes are more diverse than PRC2 complexes, which primarily function as H3K27-specific histone methyltransferases. PRC1 complexes mediate the mono-ubiquitination of histone H2A to repress transcriptional elongation for maintaining ES pluripotency[23, 24]. While protein ubiquitination and the Ub proteasome system are known to be important in maintaining pluripotency[17], much less is known about the regulatory role of deubiquitination of proteins in somatic cell reprogramming. In this respect, several members of the DUB subfamily have been shown to be involved in the maintenance of ESC pluripotency. For instance, USP family

member USP28 regulates the stability of Myc protein, which is a key factor for somatic cell reprogramming[25]. Another USP family member, USP22, acts as a transcriptional repressor of the *Sox2* locus during ESC differentiation, implicating its role in adult somatic stem cell potency[26]. Furthermore, the USP family member USP16 can remove Ub from H2AK119 and is required for ESC differentiation[14, 27]. In addition, USP7 and USP11 can regulate the ubiquitination status of PCGF2 and BMI1 in primary human fibroblast cells, leading to derepression of *INK4A*[28]. Although these studies link USP proteins to sustain pluripotency, but the role of USPs in the regulation of the components of PRC1 complexes during cellular reprogramming has never been clearly elucidated.

Using screening of shRNA specific for the DUB subfamily USPs, we identified a novel cellular reprogramming repressor gene *Usp26*, which blocked somatic cell reprogramming into

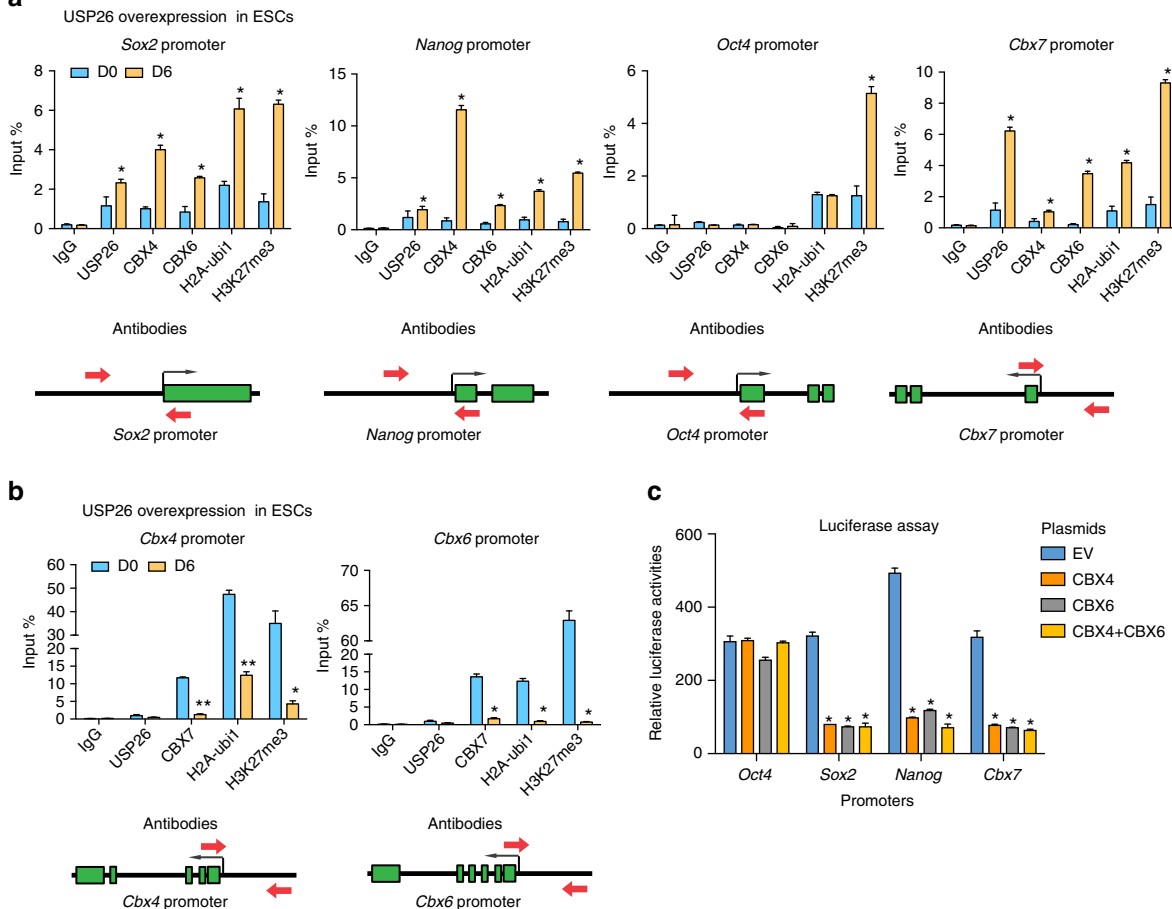

**Fig. 6** CBX4 and CBX6 proteins bind to pluripotency gene promoters and inhibit their expression. **a** ChIP–PCR analysis of *Sox2*, *Nanog*, *Oct4*, and *Cbx7* promoters in ESCs transduced with m*Usp26*, *$p < 0.5$ compared to day 0. **b** ChIP–PCR analysis of *Cbx4* and *Cbx6* promoters in ESCs transduced with m*Usp26*, *$p < 0.5$, **$p < 0.01$ compared to day 0. **c** Luciferase assay of *Oct4*, *Sox2*, *Nanog*, and *Cbx7* promoters with EV, *Cbx4*, *Cbx6*, or *Cbx4*, and *Cbx6* co-transfection, *$p < 0.5$ compared to EV. *Red arrows* indicate ChIP–PCR targets and *black arrows* indicate transcription start sites (TSSs) at the specific gene promoters. The data are presented as means ± SD from three independent experiments. **a**, **b** Unpaired two-tailed Student's *t*-test. **c** Two-way ANOVA for multiple comparisons

iPSCs and promoted differentiation by stabilizing PRC1 complexes. A major component of the canonical PRC1 complexes is CBX7, which is present within the complex while ESCs are in a pluripotent state[16, 29]. Our results demonstrate that USP26 can diminish the expression of the *Cbx7* gene and inhibit cellular reprogramming. Furthermore, the expression of CBX7 protein did not increase during ESC differentiation in contrary to the increase in CBX4 and CBX6 protein expression. Our finding is concordant with the notion that the binding of different components of the PRC1 complex acts as a switch between maintenance of pluripotency and differentiation[30]. In fact, a previous report revealed that CBX4 acted to repress the expression of *Cbx7*[22]. In line with this observation, we further demonstrated that CBX4 and CBX6 can bind to the promoter of *Cbx7*. Conversely, lesser extent of CBX7 protein binding to the promoters of *Cbx4* and *Cbx6* is observed during *Usp26*-mediated ESC differentiation and RA-induced ESC differentiation. We further identified that USP26 directly bound to CBX4 and CBX6 proteins and other components of the PRC1.2 complex, including RING1A, PCGF2, and CBX7. However, the interaction of USP26 with CBX4 and CBX6 was independent of RING1A. In differentiated cells, USP26 removed K48-linked Ub chains from CBX4 and CBX6, thus stabilizing the proteins and preventing their degradation. Henceforth, the accumulation of CBX4 and CBX6 proteins increased pluripotent gene (*Sox2* and *Nanog*) and

PRC1 gene (*Cbx7*) promoter occupancy and decreased promoter activity. The knockdown of *Cbx4* or *Cbx6* during OSKM-induced MEF reprogramming led to increased numbers of AP[+] colonies and increased pluripotent gene (*Sox2* and *Nanog*) expression, which is consistent with the improved reprogramming efficiency obtained with the knockdown of *Usp26*. However, *Usp26* shRNA combined with *Cbx4* or *Cbx6* shRNA in OSKM-induced reprogrammed MEFs did not further increase numbers of AP[+] colonies, suggesting that USP26 regulates somatic cell reprogramming through the control of CBX4 and CBX6 protein abundance, which in turn inhibit cellular reprogramming.

The knockdown of *Usp26* does not further increase *Oct4* gene expression during OSKM-induced MEF reprogramming, but the ectopic expression of *Usp26* in ESCs results in the reduction of the expression of all three pluripotency genes, suggesting that a low level of *Oct4* expression is sufficient to maintain ESC in a stable pluripotent state[31]. Correlative with the *Oct4* gene expression data, occupancy at the *Oct4* promoter by USP26, CBX4, CBX6, and H2A-ubi1 was low even during USP26-mediated ESC differentiation. However, H3K27me3 levels dramatically increased at the *Oct4* promoter, which was similar to previously published reports[32, 33]. While our data show that USP26 does not directly bind to proteins of other PRC1 complex components, such as RING1B, RYBP, PCGF1, BMI1, KDM2B, and PHC1, these complexes may still bind to the *Oct4* promoter

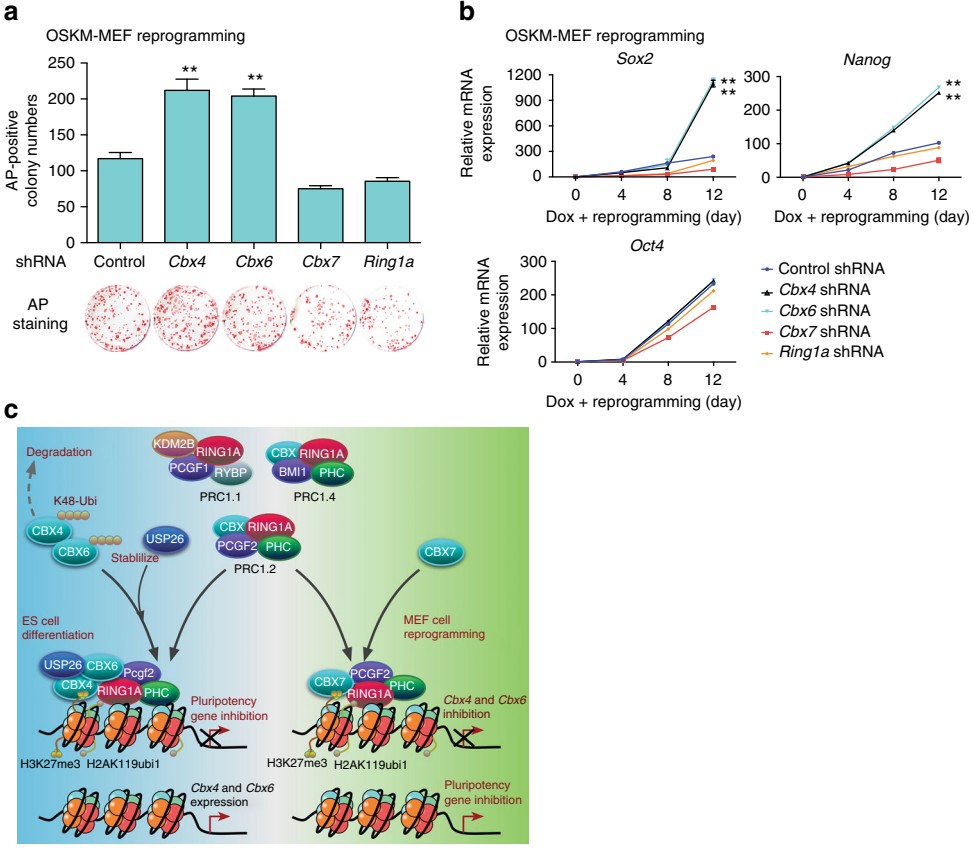

**Fig. 7** Knockdown of CBX4 and CBX6 promotes MEF reprogramming. **a** AP staining of iPSC colonies after OSKM induction in MEF cells infected with *Cbx4*, *Cbx6*, *Cbx7*, *Ring1a*, or control shRNA lentivirus, **p < 0.01 compared to control shRNA. **b** *Sox2*, *Nanog*, and *Oct4* mRNA expression levels during MEF reprogramming with *Cbx4*, *Cbx6*, *Cbx7*, *Ring1a*, or control shRNA lentivirus, **p < 0.01 compared to control shRNA. **c** Proposed role of USP26 in ESC differentiation and MEF reprogramming. USP26 removes Ub from CBX4 and CBX6 for stabilization. Accumulated CBX4 and CBX6 bind to promoters of pluripotency genes, such as *Sox2*, *Nanog*, and Polycomb genes, such as *Cbx7*. The data are presented as means ± SD from four independent experiments. **a**, **b** Two-way ANOVA for multiple comparisons

to induce USP26-independent *Oct4* gene expression. Therefore, the increased H3K27me3 levels at the *Oct4* promoter might be due to USP26-independent recruitment of PRC2.

In summary, our study identifies an important role for USP26 in cellular reprogramming by stabilizing two core components CBX4 and CBX6 of the PRC1.2 complex. Mechanistically, the availability of the accumulated CBX4 and CBX6 facilitates their binding to the promoters of the pluripotent genes and may lead to the observed inhibition of pluripotent gene expression as well as the switching of the different components of PRC1.2. As a broader perspective, the greater understanding of the regulation of these transcription factors and epigenetic modifiers in cellular reprogramming is warranted for the rapid and consistent iPSC generation.

## Methods

**MEF cell isolation**. Animal experiments were performed in accordance with an approved protocol from the Institutional Animal Care and Use committee (Houston Methodist Research Institute). Kill a pregnant OSKM mouse on 13 or 14 day post coitum by cervical dislocation. Dissect out the uterine horns and separate embryos, and then dissect head and red organs, wash in phosphate-buffered saline (PBS), and finely mince the tissue with a sterile razor blade until it becomes possible to pipette. Trypsinize each embryo with 1 ml of 0.05% trypsin/EDTA (Gibco, Invitrogen) for 15 min at 37 °C. Plate cells from three embryos in each T175 flask for 24 h (P0). Expand P0 cells till P2 or P3, and then cells were frozen for future usage.

**Cell culture**. Human embryonic kidney (HEK) 293T cells were obtained from ATCC and maintained in Dulbecco's modified Eagle's medium (DMEM; Hyclone) with 10% fetal bovine serum (Valley Biomedical) and 1% antibiotic–antimycotic

solution (Gibco). ESCs were maintained in ES culture medium (DMEM supplemented with 10% Knockout serum, 2 mM L-glutamine, 100 μM non-essential amino acids (Gibco), 0.1 mM ß-mercaptoethanol, and 50 ng ml$^{-1}$ LIF). For Usp26-induced differentiation, ES cells were transduced with lentiviruses expressing Dox-inducible GFP-tagged Usp26 or empty vector (Day −3). After ESCs formed colonies, 0.1 μg ml$^{-1}$ Dox was added for pre-selection (Day −1). Day 0 was defined as the day when pre-selected GFP-positive colonies were cultured in iSF1 medium with high-dose Dox (2 μg ml$^{-1}$), and individual colonies were tracked and taken pictures on days 1, 2, 3, and 4. For RA-induced differentiation, *Usp26* KO, *Cbx4* KO, *Cbx6* KO, or WT ES cells were cultured in ES differentiation medium (DMEM supplemented with 10% Knockout serum, 2 mM L-glutamine, 100 μM non-essential amino acids (Gibco), and 0.1 mM ß-mercaptoethanol) with 1 μM RA. Individual colonies were tracked and photographed over 8 days.

**Generation and reprogramming efficiency evaluation of iPSCs**. Mouse iPSCs were generated from MEFs with some minor modifications[2]. Tet-O-inducible OSKM MEFs were used to generate iPSCs. OSKM transgenic MEF cells were transduced with GIPZ lentivirus-based shRNAs targeting USP family members, *Cbx4*, *Cbx6*, or *Cbx4*, and *Cbx6*. These cells were selected with 2 μg ml$^{-1}$ puromycin for 2 days. Before reprogramming, $3 \times 10^5$ feeder cells (irradiated MEFs) per well were seeded into six-well plates previously coated with 0.1% gelatin. Thousand puromycin-selected OSKM transgenic MEF cells were reseeded onto feeder cells in six-well plates. The next day, modified iSF1 medium (DMEM supplemented with 10% Knockout serum, 2 mM L-glutamine, 100 μM non-essential amino acids (Gibco), 0.1 mM ß-mercaptoethanol, 50 ng ml$^{-1}$ LIF, 5 μg ml$^{-1}$ CHIR99021, and 2.5 μg/ml PD0325901) containing 2 μg ml$^{-1}$ Dox was added and replenished every day. The efficiency of iPSC formation was calculated based on the number of AP$^+$ iPSC colonies. The colonies were stained for AP activity on day 12 with AP detection Red Substrate kit (Vector).

**Teratoma assay**. iPSCs ($1 \times 10^6$) were injected subcutaneously into the skin on the dorsal rear flank of five severe combined immunodeficiency (SCID) mice (6 week, female). Four weeks after injection, mice were killed and tumors were removed and

fixed in formalin for 24 h. These tumors were imbedded in paraffin, sectioned, and stained with hematoxylin and eosin for histological analysis.

**Immunostaining.** To fully characterize iPSCs, these cells were fixed with methanol for 20 min at −20 °C, and nonspecific receptors were blocked with 10% normal goat serum. Cells in culture plates or chamber slides were fixed for 20 min at −20 °C with methanol, and nonspecific receptors were blocked with 10% normal goat serum. Oct4, SSEA1, and Nanog were stained with specific antibodies (anti-Oct4, 1:500, Santa Cruz Biotechnology SC-5279; anti-Nanog, 1:500, Santa Cruz Biotechnology SC-293121; and anti-Dppa4, 1:500, R&D AF3674; anti-SSEA1, 1:1000, Invitrogen MA1-022), followed by goat anti-mouse antibody-conjugated Texas Red (1:2000, Invitrogen T-862). Nuclei were stained with 4,6-diamidino-2-phenylindole (DAPI; Invitrogen). Immunofluorescence staining was visualized, and cells were photographed with an Olympus 1X71S1F fluorescence microscope.

**qPCR and ChIP–PCR.** Total RNA was isolated using an RNeasy Mini Kit (Qiagen). Reverse transcription was performed using SuperScript III reverse transcriptase (Invitrogen). Real-time PCR (qPCR) was performed using a StepOne$^{TM}$ Real-Time PCR Systems and Power SYBR® PCR Master Mix (Life Technologies). qPCR primers are listed in Supplementary Table 1.

Mouse ESCs were grown to 80–90% confluence and were chemically crosslinked by the addition of fresh formaldehyde solution (37%) to a final concentration of 1% for 10 min at room temperature. Cells were rinsed twice with cold PBS followed by addition of 2 M glycine to stop crosslinking and were collected using a silicon scraper. Cells were lysed and sonicated to solubilize and shear crosslinked DNA, with a minor modification. Briefly, we used an ultrasonic liquid processor (Misonix) and sonicated at an amplification of 4 for 12 × 10 s pulses (30 s pause between pulses) at 4 °C. The resulting whole-cell extract was pre-cleared with 50 μl protein A/G beads, 10 μl IgG, 10 μl 5% bovine serum albumin (BSA), and 5 μg of sheared salmon sperm DNA for each sample. After centrifugation, 20% of the supernatant was incubated overnight at 4 °C with 30 μl of Protein A/G agarose beads and 3 μg of the appropriate antibodies, 1 μl BSA (5%), and 25 μg of sheared salmon sperm DNA. Beads were washed four times with ChIP buffer (0.1% SDS, 1% Triton X-100, 2 mM EDTA (pH 8.0), 150 mM NaCl, and 20 mM Tris-HCl (pH 8.0)) and once with tris-EDTA (TE) containing 1 mM dithiothreitol (DTT). Bound complexes were eluted from the beads, and crosslinking was reversed by overnight incubation at 65 °C in reverse crosslink buffer (1% SDS, 100 mM NaHCO$_3$, 1 μg ml$^{-1}$ RNase A, and 500 mM NaCl). Whole-cell extract DNA was also treated for reverse crosslinking. Immunoprecipitated DNA and whole-cell extract DNA were purified by Zymoclean PCR purification kit (Zymo). ChIP–PCR primers are listed in Supplementary Table 1. Antibodies used for ChIP–PCR: anti-USP26, 1:100, Abcam ab101650; anti-RING1A, 1:100, Cell Signaling 13069; anti-CBX4, 1:100, Santa Cruz Biotechnology sc-130822; anti-CBX6, 1:100, Santa Cruz Biotechnology sc-86355; anti-CBX7, 1:100, Santa Cruz Biotechnology sc-376274; anti-H2Aui1, 1:100, Cell Signaling 8240; anti-H3K27me3, 1:200, Abcam ab6002; anti-H3K4me3, 1:200, Abcam ab8580; anti-PCGF2, 1:100, Santa Cruz Biotechnology sc-130415.

**IP and western blotting.** For IP, whole-cell extracts were prepared after transfection or stimulation with appropriate ligands, followed by incubation overnight with the appropriate antibodies plus protein A/G beads (Pierce). Beads were washed five times with low-salt lysis buffer, and immunoprecipitates were eluted with 4× SDS loading buffer and resolved by SDS-PAGE. Proteins were transferred to nitrocellulose membranes (Bio-Rad) followed by further incubation with the appropriate antibodies. Luminata Crescendo Western HRP substrate (Millipore) was used for protein detection. For endogenous IP, mouse ESC nuclear extracts (150 μl for each IP) were immunoprecipitated with appropriate antibodies (3 μg for each IP) followed by western blotting or mass spectrometry analysis. As controls, either rabbit or mouse IgG antibodies were used. For interaction studies, 293T cells were co-transfected with plasmids encoding various potential USP26 proteins in different combinations using the Lipofectamine 2000 (Invitrogen) method. At 48 h after transfection, cells were lysed in cell lysis buffer (50 mM Tris-HCl (pH 8.0), 150 mM NaCl, 1 mM EDTA, 1% Nonidet P-40, 2 mM MgCl$_2$, 8 U benzonase, and 10% glycerol with protease inhibitor mixture) for 30 min, followed by IP of cell lysates with anti-FLAG M2 antibody (1:2000, Sigma F3165). Next, immunocomplex pulled down by anti-FLAG M2 antibody was subjected to western blotting with anti-RING1A, CBX4, CBX6, CBX7, and PCGF2 antibodies. As controls, whole-cell extracts were fractionated by SDS-PAGE, followed by immunoblotting with specific or anti-FLAG antibodies. Antibodies used for western blotting, anti-USP26, 1:1000, Abcam ab101650; anti-RING1A, 1:1000, Cell Signaling 13069; anti-CBX4, 1:1000, Santa Cruz Biotechnology sc-130822; anti-CBX6, 1:1000, Santa Cruz Biotechnology sc-86355; anti-CBX7, 1:1000, Santa Cruz Biotechnology sc-376274; anti-PCGF2, 1:1000, Santa Cruz Biotechnology sc-130415; anti-H2A, 1:1000, Abcam ab18255; anti-Ub, 1:1000, Santa Cruz Biotechnology sc-271289. Original blots were provided in Supplementary Fig. 7.

**Plasmids constructs.** Full-length mouse *Usp*, *Cbx4*, and *Cbx6* cDNA were obtained from MEF cDNA by two-step PCR and cloned into pcDNA3.1 with HA tag sequence. A similar strategy was used to clone human USP26. The C-terminal GFP-tagged m*Usp26*, m*Cbx4*, and m*Cbx6* cassettes were amplified by overlapping PCR and cloned into plTet-O lentiviral vector through BP and LR reactions of Gateway cloning system (Invitrogen). plTet-O-mUsp26-GFP, plTet-O-mCbx4-GFP, plTet-O-mCbx6-GFP, pcDNA-HA-hUSP26, and pcDNA-HA-mUsp26 were sequenced to verify the correct DNA sequence and their open reading frames.

**Lentivirus production and transduction.** One day before transfection, HEK293T cells were seeded at 50% confluency in 15 cm dishes. Cells were transfected the next day at 80–90% confluency. For each flask, 20 μg of plasmid containing the vector of interest, 10 μg of VSV-G, and 15 μg of Δ8.9 were transfected using calcium transfection methods. Five hours (h) after transfection, the media was changed. Virus supernatant was harvested at 48 h post transfection, filtered with a 0.45 μm polyvinylidene fluroride (PVDF) filter (Millipore), and centrifuged at 25,000 × g for 2 h. Lentiviral pellet was resuspended with 1 ml target cell medium. MEFs were cultured in DMEM supplemented with 10% Knockout serum, 2 mM L-*glutamine*, 100 μM non-essential amino acids (Gibco), and ß-Mercaptoethanol, and passaged every other day at a 1:4 ratios. Cells were transduced with lentivirus via spinfection in six-well plates. One thousand cells in 2 ml of media supplemented with 8 mg ml$^{-1}$ polybrene (Sigma) were added to each well, supplemented with 10 μl lentiviral supernatant. Medium was refreshed on day 2, and cells were passaged every other day starting on day 4 after replating.

**CRISPR knockout.** Short guide RNAs (sgRNAs) were designed using the CRISPRtool (http://crispr.mit.edu) to minimize potential off-target effects. sgRNA sequences and genomic primers are listed in Supplementary Table 1. Corresponding oligonucleotides were ordered (IDT) and subcloned into the LentiCRISPR plasmid (Addgene), expressing a human codon-optimized SpCas9, guide RNA, and puromycin expression plasmid, following a previously published protocol[34]. Specific sgRNA lentiviruses were packaged as described in the Methods section. One milliliter lentiviral supernatant was added into 1 × 10$^6$ 293T cells or 1 × 10$^5$ MEFs in six-well plates. After 48 h, 2 or 4 μM puromycin was used for selection. Puromycin-selected 293T or MEF cells (Fig. 4c and Supplementary Fig. 3b–e) or MEFs (Supplementary Fig. 2c, d) were expanded for western blot assay.

**TIDE assay.** Genomic regions surrounding sgRNA-targeted sites were amplified by PCR. TIDE assay[35] primers are listed in Supplementary Table 1. PCR products were purified using the Zymoclean™ Gel DNA Recovery Kits (Zymo) and sequenced. Sequencing results were analyzed with TIDE web tool (https://tide.nki.nl).

**Deubiquitinase activity assay.** FLAG-USP26 (1 mM) proteins were incubated with 100 ng of poly-linked Ub chains (K48 or K29, BIOMOL) for 3 h at 37 °C in 50 mM Tris (pH 8.0) and 1 mM DTT, separated by SDS-PAGE, and analyzed by anti-Ub immunoblotting (1:1000 dilution; Santa Cruz Biotechnology).

**Luciferase assay.** 293T cells (8 × 10$^4$ cells) were transduced with reporter (pLenti-Luc-OCT4, SOX2, NANOG, or CBX7) lentiviral supernatants and then transfected with 100 ng HA-USP26, HA-CBX4, or HA-CBX6. Cell lysate was prepared following harvesting cells 24 h after transfection, and reporter activity was measured with the Dual Luciferase Assay (Promega).

**Statistical analysis.** Significant differences between groups were assessed with two-way ANOVA test or two-tailed Student's *t*-test. Values of $p < 0.05$ were considered statistically significant.

**Data availability.** The data sets generated during and analyzed during the current study are available from the corresponding author on reasonable request.

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

## Acknowledgements

This work was in part supported by grants from the NCI, NIH (R01CA090327 and R01CA101795), Cancer Prevention and Research Institute of Texas (CPRIT), National Natural Science Foundation of China (No. 81572766), and Dottie and Jimmy C. Adair Myelodysplastic Syndrome Research and Treatment Fund. W.L. was supported in part by Xiangya Hospital, Xiangya School of Medicine, Central South University, China. We would like to thank Dr. Jana Burchfield for critical reading of this manuscript.

## Author contributions

Conceptualization: B.N. and R.-F.W.; methodology: W.Z., C.Q., Q.L., P.L., and W.L.; writing—original draft: B.N.; writing—review and editing: R.-F.W.; funding acquisition: R.-F.W.

## Additional information

**Competing interests:** The authors declare no competing financial interests.

