## [Peer Review File · Nature Communications]

Reviewers' comments:

Reviewer #1 (Remarks to the Author):

This manuscript touches upon a very interesting topic both from a basic as well as translation point of view that is the efficiency of cellular reprogramming.

The authors have provided a strong evidence to justify that Usp26 plays important roles in inhibiting reprogramming through stabilization of CBX4/6 members of the polycomb repressive complex 1 and silencing of genes such as Nanog and Sox2.

Some parts of the study need clarification before the manuscript is further considered for publication:

Pg 5. "Forty-eight mouse Usp family member shRNAs or reference shRNA were transduced into inducible Oct4-Sox2-Klf4-cMyc (OSKM) transgenic MEFs."

-How many members does USP family consist of? The authors should explain/describe why they chose to study 48 of them.

-There are some other USP proteins that might also affect reprogramming efficiency based on the data presented here (such as Usp30, Fig. 1b). The authors should comment on this at this point in the text.

Pg 6. "Usp26 mRNA sharply decreased during Dox-induced OSKM-mediated MEF reprogramming and increased during ESC differentiation (Fig. 2c, d)."

-what is the mechanism of Usp26 repression in ESC? Is Usp26 locus bound by Cbx7 in ESC?

Pg 7. "To determine whether USP26 targets subunits of the PRC1 complexes, immunoprecipitation (IP) was performed to screen for protein-protein interactions.

... Only undifferentiated ESCs expressed CBX7 without expression of USP26, suggesting that the interaction between USP26 and CBX7 observed in 293T cells is non-specific."

-As 293T cell physiology is different than the one of ES cells and protein-protein interactions might depend on post-translational modifications, the authors should repeat those IPs in ES cells. There are good antibodies for the endogenous proteins for some of those cases.

-The authors should also include ensure the interactions are direct and not DNA-mediated by including factors such as benzonase into their IP buffer.

. Pg. 7. "Upon Dox-induced OSKM-mediated MEF reprogramming, Cbx7 gene expression was increased when Usp26 was knocked down (Fig. 3a) and decreased when Usp26 was overexpressed (Fig. 3b). These data suggest that the PRC1 component CBX7 is a potential target of USP26."

-How does Usp26 control Cbx7? Can this be through modulation of the H2A/B ubiquitylation levels or it is related to stabilization of Cbx4/6 only? Figure 6a shows that USP26 binding is particularly enriched in CBX7 locus compared to SOX2 and Nanog.

Pg 9. "CBX6 protein levels were also dramatically increased upon addition of the proteasome inhibitors, MG132 or Lactacystin (Supplementary Fig. S5b), in cells overexpressing USP26, suggesting that USP26 may prevent proteasomal degradation of CBX4 and CBX6."

-Actin loading control is saturated in this figure. I would repeat this by loading less protein so actin bands are not saturated and the results cleaner.

pg 11. "After 12 days of Dox treatment, knockdown of Cbx4 or Cbx6, but not Cbx7 or Ring1a, led to an increased number of AP+ colonies compared with the control shRNA group (Fig. 7a)."

-The authors should try compound knockdown of both Cbx4 and 6 to monitor additive effect. Along those lines (and based on the results in figure 6c) do both of Cbx4 and Cbx6 bind simultaneously on the same promoter and control genes such as Sox2 and Nanog or one of the 2 proteins is sufficient to downregulate these loci?

Reviewer #2 (Remarks to the Author):

In this manuscript the authors demonstrated that Usp28 knockdown enhances reprogramming efficiency and also delays RA-induced ESC differentiation. Conversely, over expression of Usp28 decreases reprogramming efficiency and also induces ESC differentiation. Mechanistically, USP28 physically interacts with PRC1 components CBX4 and CBX6, deubiquitinates, and bind to the Sox2, Nanog and Cbx7 promoters together with CBX4 and CBX6.

Although both Usp28 knockdown and overexpression seem to show very strong phenotype, I found the experimental settings need to be explained in more detail. For example, control samples make 100 ALP+ colonies from 1000 MEFs in their reprogramming system. If all ALP+ colonies are real iPSC colonies, reprogramming efficiency is 10%, which is very high. The authors need to use more stringent methods to evaluate reprogramming efficiency (such as Nanog, Dppa4 immunofluorescence) and also describe how the system is made in detail. Similarly, the Usp28 overexpression experiment (Fig 2f) is also not clear how it was done. The legend says GFP-tagged mUsp26 overexpression, but then how the authors can maintain the green colonies shown on day 0? How many shRNAs have been tested, and what are the sequences? I have several similar questions as below. I also point a few data interpretation, which I found too rough.

Fig 1f, Nanog IF. It does not look nuclear staining.

Regarding Fig 2c, the authors described 'Usp26 mRNA sharply decreased during Dox-induced OSKM-mediated MEF reprogramming' This looks like a gradual decrease, rather than a sharp decrease.

Regarding Fig 2f, the authors described 'infection of ESCs with GFP-tagged mouse (m) Usp26 led to ESC differentiation in an RA-independent manner (Fig. 2f), suggesting that USP26 may be downstream of RA signaling.' This is too rough statement. ESC differentiation can be caused without RA.

Supplementary Fig. S2c. How the Usp26 KO MEF lines are generated? MEFs usually get senescence before forming colonies

Supplementary Fig. S3 Circos plots, it is not possible to understand without explanations.

Fig 3b, Control endogenous Oct4. Why the value is only 20? Fig 3a control has 200.

Based in Fig 3, the authors described 'These data suggest that Sox2 and Nanog gene expression is regulated by Usp26 during Dox-induced OSKM-mediated MEF reprogramming.' 'These data suggest that the PRC1 component CBX7 is a potential target of USP26.' Any other pluripotency genes can show similar expression pattern. This data itself does not suggest direct regulation of Sox2, Nanog, Cbx7 expression by Usp28.

Fig 3c. Was USP26 overexpression before RT-PCR performed with lentivirus? If so, is which time point day0?

About Fig 4b, the authors stated 'Only undifferentiated ESCs expressed CBX7 without expression of USP26, suggesting that the interaction between USP26 and CBX7 observed in 293T cells is non-specific.' This does not need to be called 'non-specific'. If there are cells which express endogenous Cbx7 and USP26, they might interact.

Related Fig 5b, what the authors need to show is if Usp26 KD decreases CBX4 and CBX6.

Fig 5c, CBX6 up-regulation seems to take a long time, and does not look like a direct consequence of exogenous Usp26 expression. During RA-induced ESC differentiation where Usp26 expressing

goes up, do CBX4 and 6 expression level increase? If so, this could be a result of USP26-induced differentiation.

Supplementary Fig. S5b. Proteasome inhibitors do not seem to increase CBX4 and CBX6 in 293T cells in the absence of exogenous USP28. It suggests that ubiquitination-mediated degradation is not active in this cell line. Why the proteasome inhibitors' effects become obvious only when exogenous USP28 is expressed? I expect that exogenous USP28 remove ubiquitin, stabilize CBX4, CBX6, therefore the proteasome inhibitors have less impact on the CBX4, CBX6 protein level. Anyway, this experiment needs to be done in MEFs.

Supplementary Fig S6. It is important to show protein levels of CBX4, 6 in RA differentiation, since USP26 controls CBX4, 6 levels at the post translational level.

Authors stated 'Increased Cbx4 and Cbx6 mRNA was also observed in Usp26-induced or RA-induced ESC differentiation (Supplementary Fig. S6b, c)' S6c seems to be a wrong figure.

Supplementary Fig. S7b. This is also not clear how day0 colonies are maintained in the presence of the transgenes.

Discussion 'Usp26 overexpression in OSKM-induced reprogrammed MEFs did not significantly increase Cbx4 or Cbx6 gene expression, but increased CBX4 and CBX6 protein expression, suggesting that USP26-mediated deubiquitination of CBX4 and CBX6 is the predominant mechanism for CBX4 and CBX6 accumulation during somatic cell reprogramming.' I am not sure which data showed 'increased CBX4 and CBX6 protein expression' during reprogramming.

Response to Reviewer 1:

- Comment:* Pg 5. "Forty-eight mouse Usp family member shRNAs or reference shRNA were transduced into inducible Oct4-Sox2-Klf4-cMyc (OSKM) transgenic MEFs."? How many members does USP family consist of? The authors should explain/describe why they chose to study 48 of them.

Response: There are total 54 members of mouse USP family. In this study, to knockdown USP family members we used commercially available shRNA library (Dharmacon) that consists of 48 shRNAs available for mouse family members. We tested at least 2 individual shRNAs of each gene and chose shRNA with high knockdown efficiency (>70%) for screening. In the revised version, we added the information regarding the 48 USP used in our initial screening on Page 5, Line 9- Line 12. All shRNA target sequences were listed in Supplementary Table S1 (Pages 49-51).

- Comment:* There are some other USP proteins that might also affect reprogramming efficiency based on the data presented here (such as Usp30, Fig. 1b). The authors should comment on this at this point in the text.

Response: Thanks for this excellent suggestion. Reprogramming efficiency of Usp30 or Usp52 knockdown MEF cells was also slightly increased (less than 2-fold) compare to control, however, the effect was not as strong as Usp26 or Usp20 knockdown. We have added the descriptions of Usp30 and Usp52 in the main text (Page 5, Lines 17-18; Lines 20-22)

- Comment:* Pg 6. "Usp26 mRNA sharply decreased during Dox-induced OSKM-mediated MEF reprogramming and increased during ESC differentiation (Fig. 2c, d)." What is the mechanism of Usp26 repression in ESC? Is Usp26 locus bound by Cbx7 in ESC?

Response: We examined the promoter binding activity of Usp26 by PRC1 using ChIP qPCR during RA-induced ESC differentiation. Our results showed that occupation of PRC1 components PCGF2 and Cbx7 was significantly decreased in the USP26 promoter (Figure S6d), also, more H2Aubi1

was observed in *Usp26* promoter. Thus, Cbx7-containing PRC1 complex is potential repressor of *Usp26*. We have incorporated these new data in the text on Page 11 Line 10-14.

Figure S6d. ChIP-PCR analysis of binding of PRC1 complex proteins to the *Usp26* promoters in ESC differentiation upon LIF withdrawal and treatment with RA

4. *Comment:* Pg 7. "To determine whether USP26 targets subunits of the PRC1 complexes, immunoprecipitation (IP) was performed to screen for protein-protein interactions.

... Only undifferentiated ESCs expressed CBX7 without expression of USP26, suggesting that the interaction between USP26 and CBX7 observed in 293T cells is non-specific. "

-As 293T cell physiology is different than the one of ES cells and protein-protein interactions might depend on post-translational modifications, the authors should repeat those IPs in ES cells. There are good antibodies for the endogenous proteins for some of those cases.

-The authors should also include ensure the interactions are direct and not DNA-mediated by including factors such as benzonase into their IP buffer.

Response: Thanks for these excellent suggestions. We have repeated IP experiments in mouse ES

cells with suggested IP buffer (benzonase with 2 mM MgCl₂). Since only undifferentiated ESCs expressed CBX7 without expression of USP26, endogenous interactions between USP26 and CBX4 or CBX6, but not CBX7, were observed in differentiating ESCs (Fig. 4b). We have included these new findings in the text on **Page 8 Lines 8-12 and Methods section, Page 18 Lines 15-17,**

5. *Comment:* Pg. 7. "Upon Dox-induced OSKM-mediated MEF reprogramming, Cbx7 gene expression was increased when Usp26 was knocked down (Fig. 3a) and decreased when Usp26 was overexpressed (Fig. 3b). These data suggest that the PRC1 component CBX7 is a potential target of USP26" ?

-How does Usp26 control Cbx7? Can this be through modulation of the H2A/B ubiquitylation levels or it is related to stabilization of Cbx4/6 only? Figure 6a shows that USP26 binding is particularly enriched in CBX7 locus compared to SOX2 and Nanog.

Response: Our results showed that USP26 binding is particularly enriched in CBX7 locus compared to SOX2 and Nanog (Figure 6a). To study whether Usp26 can regulate Cbx7 directly or co-operate with Cbx4 and Cbx6, we performed luciferases assay to analyze the promoter activities of Oct4, Sox2, Nanog and Cbx7. The results showed that overexpression of Usp26 alone could not

inhibit promoter activities of these pluripotency genes. However, when Usp26 was co-expressed with either Cbx4 or Cbx6, significantly decreased promoter activities of Sox2, Nanog and Cbx7 were observed (Figure S6g). These data suggest that Cbx4 or Cbx6 accumulation in the Cbx7 promoter is essential for Cbx7 suppression. (see revised text on Page 11, Lines 9-15).

Figure 6a. ChIP-PCR analysis Cbx7 promoter in ESCs transduced with mUsp26 (* $p < 0.01$ compared to day 0).

Figure S6g. Luciferase assay of Oct4, Sox2, Nanog, and Cbx7 promoters with empty vector (EV), Usp26, or Usp26 and Cbx4 or Cbx6 co-transfection (** p<0.01 compared to EV).

6. *Comment:* Pg 9. "CBX6 protein levels were also dramatically increased upon addition of the proteasome inhibitors, MG132 or Lactacystin (Supplementary Fig. S5b), in cells overexpressing USP26, suggesting that USP26 may prevent proteasomal degradation of CBX4 and CBX6. " Actin loading control is saturated in this figure. I would repeat this by loading less protein so actin bands are not saturated and the results cleaner.

Response: We have repeated this experiment in 293T cells, and loading amount of protein was also reduced to avoid saturation, as reviewer suggested. In the revised Figure S5f, it is clear that actin bands are not saturated while CBX4 and CBX6 protein levels were dramatically increased upon the addition of the proteasome inhibitors, MG132 or Lactacystin in 293T cells, thus suggesting that USP26 may prevent proteasomal degradation of CBX4 and CBX6 in 293T cells (see amended text on Page 9 Lines 7- 8)

Figure S5e. 293T cells were treated with DMSO, MG132, Lactacystin, or PR619. Specific proteins were analyzed by western blotting using anti-CBX4, anti-CBX6 antibodies, and anti-USP26 antibodies. β -actin was used as a loading control.

7. *Comment:* pg 11. "After 12 days of Dox treatment, knockdown of Cbx4 or Cbx6, but not Cbx7 or Ring1a, led to an increased number of AP⁺ colonies compared with the control shRNA group (Fig. 7a)" ?

-The authors should try compound knockdown of both Cbx4 and 6 to monitor additive effect. Along those lines (and based on the results in figure 6c) do both of Cbx4 and Cbx6 bind simultaneously on the same promoter and control genes such as Sox2 and Nanog or one of the 2 proteins is sufficient to downregulate these loci?

Response: We have also analyzed Sox2, Nanog and Cbx7 promoters in Cbx4 and/ or Cbx6 knockdown ES cells. ChIP-qPCR results showed that Cbx4 or Cbx6 alone bound more in Sox2, Nanog and Cbx7 promoters (Supplementary Fig. S6e&f). Usp26 and H2Aubi1 also were recruited to these promoters (Supplementary Fig. S6e&f), suggesting repression of Sox2, Nanog and Cbx7 expression. However, there is no promoter binding observed when both Cbx4 and Cbx6 were knocked down in ES cells (Supplementary Fig. S6e&f). Cbx4 or Cbx6 overexpression dramatically

decreased promoter activities of Sox2, Nanog and Cbx7 (Fig. 6c). These results indicate that Cbx4 or Cbx6 alone is sufficient for downregulating Sox2, Nanog or Cbx7 (Page 11 Line 1-5).

e Usp26 antibody pull-down

f H2Aubi1 antibody pull-down

Figure S6e-g. ChIP-PCR analysis of *Sox2*, *Nanog* and *Cbx7* promoters in Cbx4 shRNA (e), Cbx6 shRNA (f) or Cbx4 and Cbx6 shRNA (g) transduced ESCs, which were cultured upon LIF withdrawal and treatment with RA (* p<0.05, ** p<0.01 compared to day 0).

Figure 6c. Luciferase assay of *Oct4*, *Sox2*, *Nanog*, and *Cbx7* promoters with empty vector (EV), *Cbx4*, *Cbx6*, or *Cbx4* and *Cbx6* co-transfection (* $p < 0.01$ compared to EV).

Reviewer 2

1. *Comment:* The authors need to use more stringent methods to evaluate reprogramming efficiency (such as *Nanog*, *Dppa4* immunofluorescence) and also describe how the system is made in detail.

Response: Thanks for this excellent suggestion. We have re-evaluated reprogramming efficiency of *Usp26* with both AP staining and *Nanog* immunofluorescence of iPSC colonies (Supplementary Fig.1b & c).

Reprogramming system and efficiency: Mouse iPSCs were generated from MEFs, as previously described⁴. Tet-O inducible *Oct4*-*Sox2*-*Klf4*-*cMyc* (OSKM) MEFs were used to generate iPSCs by treating MEFs with doxycycline (Dox) in mESC medium. The initial cell number of seeded OSKM transgenic MEF cells (1000) were transduced with lentivirus-based shRNAs specific for mouse USP family members and then reseeded on irradiated feeder cells at the desired density. The next day, modified iSF1 medium (DMEM supplemented with 10% Knockout serum, 2 mM L-Glutamine, 100 μ M Non-Essential Amino acids (Gibco), 0.1 mM β -Mercaptoethanol, 50 ng/ml LIF, 5 μ g/ml CHIR99021, 2.5 μ g/ml PD0325901) containing 2 μ g/ml Dox was added and replenished every day.

The efficiency of iPSC formation was calculated based on the number of AP+ iPSC colonies and Nanog+ iPSC colonies. The colonies were stained for AP activity on days 12 with AP detection Red Substrate kit (Vector). For Nanog immunofluorescence, iPSCs were fixed for 20 min at -20°C with methanol, and nonspecific receptors were blocked with 10% normal goat serum. Nanog was stained with anti-Nanog antibody (Santa Cruz Biotechnology); followed by goat anti-mouse antibody conjugated Texas Red (Invitrogen). Nuclei were stained with DAPI (4,6-diamidino-2-phenylindole; Invitrogen). Immunofluorescence staining was visualized, and Nanog + colonies were counted with an Olympus 1X71S1F fluorescence microscope (Methods section, Page 16 line 10 – Page17 line 2).

Figure S1b. Immunofluorescence microscopic images of Nanog in Usp26 knockdown iPSCs

c OSKM-MEF reprogramming

Figure S1c. Quantification of Nanog⁺ colonies after 12 days of OSKM induction in MEFs transduced with control or Usp26 shRNA (** p<0.01 compared to control shRNA).

2. *Comment:* Similarly, the Usp26 overexpression experiment (Fig 2f) is also not clear how it was done. The legend says GFP-tagged mUsp26 overexpression, but then how the authors can maintain the green colonies shown on day 0?

Response. We have added the detailed description of this method in Methods section, (Page 18, line 21- Page 19, line 2). The C terminal GFP-tagged mUsp26 cassette was amplified by overlapping PCR and cloned into pTet-O lentiviral vector through BP and LR reactions of Gateway cloning system. mUsp-GFP lentivirus production was described in Methods section. For lentiviral transduction, mouse ES cells were seeded in 0.1% gelatin coated 6-well plate and were transduced with lentivirus via spinfection. One thousand cells in 2 ml of media supplemented with 8 mg/ml polybrene (Sigma) were added to each well, supplemented with 50 μ l lentiviral supernatants. mESCs were maintained in ES medium (DMEM supplemented with 10% knockout serum, 2 mM L-Glutamine, 100 μ M Non-Essential amino acids (Gibco), 0.1 mM β -Mercaptoethanol, and leukemia inhibitory factor (LIF) 50 ng/ml (Santa Cruz Biotechnology)) after 4 hours transduction. After 2 day

culture, mESC formed small colonies in plates, then the medium was changed with fresh medium plus 2µg/ml doxycycline to induce Usp26 expression. So we define this day Usp26 inducible expression by Dox as Day 0, and green colonies are maintained.

3. *Comment:* How many shRNAs have been tested, and what are the sequences?

Response: At least 2 shRNA were tested for each gene, and shRNAs with high knockdown efficiency (>70%) were selected for target gene knockdown. We list all used shRNA sequences in Supplementary Table S1, Page 53 Line 37- Page 55 Line 13.

4. *Comment:* Fig 1f, Nanog IF. It does not look nuclear staining.

Response: Thanks for this comment. We have repeated experiments with different antibodies for Figure 1f, on Page 25

Figure 1f. Immunofluorescence microscopic images of Oct4, Nanog, and Dppa4 in Usp26 knockdown iPSCs. iPSC colonies were fixed, blocked and stained with specific antibodies, followed by goat anti-mouse antibody conjugated Texas Red. Nuclei were stained with DAPI. Scale bar, 100 μ M.

5. *Comment:* Regarding Fig 2c, the authors described 'Usp26 mRNA sharply decreased during Dox-induced OSKM-mediated MEF reprogramming'. This looks like a gradual decrease, rather than a sharp decrease.

Response: We have revised this description to: “gradually decrease” on Page 6, Line 13-14

6. *Comment:* Regarding Fig 2f, the authors described 'infection of ESCs with GFP-tagged mouse (m)

Usp26 led to ESC differentiation in an RA-independent manner (Fig. 2f), suggesting that USP26 may be downstream of RA signaling.' This is too rough statement. ESC differentiation can be caused without RA.

Response: We appreciate this comment on RA signaling and Usp26 regulation. Usp26 overexpression causes ES differentiation, as RA-induced ESC differentiation, so we have removed this statement.

7. *Comment:* Supplementary Fig. S2c. How the Usp26 KO MEF lines are generated? MEFs usually get senescence before forming colonies

Response: The description of this USP26 KO MEF is not clear enough. We used LentiCRIPSR-v1 vector, which is a puromycin expressing vector to knockout Usp26, 4 μ M puromycin was used for selection. (Supplementary Methods section, Page 52 Line 23- Page58 Line 3).

8. *Comment:* Supplementary Fig. S3 Circos plots, it is not possible to understand without explanations.

Response: We have added more detailed information to explain Circos plots in supplementary Fig S3.

Circos plots display contributions of pluripotency genes for MEF cell reprogramming after Usp26 knockdown. Left half of the plot (outside ring) shows days after reprogramming start. The right half of the plot shows pluripotency genes. Each arc shows links between pluripotency genes and reprogramming days. In this diagram, Cbx7, Sox2 and Nanog (purple segments) show the most significant change in both Day 4 and day 12. (Page 40 line 2–7).

9. *Comment:* Fig 3b, Control endogenous Oct4. Why the value is only 20? Fig 3a control has 200.

Response: In figure 3b the value should be 300, and Y axis scales should be 0, 100, 200, 300 and 400. (See Page 29).

Figure 3b-Oct4. Real-time PCR (qPCR) analysis of mouse Oct4 mRNA expression in ESCs transfected with mouse Usp26 or empty vector (EV).

10. *Comment:* Based in Fig 3, the authors described 'These data suggest that Sox2 and Nanog gene expression is regulated by Usp26 during Dox-induced OSKM-mediated MEF reprogramming.' 'These data suggest that the PRC1 component CBX7 is a potential target of USP26.' Any other pluripotency genes can show similar expression pattern. This data itself does not suggest direct regulation of Sox2, Nanog, Cbx7 expression by Usp26

Response: Based on Circos analysis (Supplementary Fig S3a), Cbx7, Nanog and Sox2 are noted to be the most significantly changed genes in MEFs during reprogramming with Usp26 knockdown. (Page 7, Line 18-20).

11. *Comment:* Fig 3c. Was USP26 overexpression before RT-PCR performed with lentivirus? If so, is which time point day0?

Response: As we described above, Usp26 overexpression in ESCs is controlled by Tet-O inducible system. For pluripotency genes mRNA level analysis with q-PCR, ES colonies morphologies are not essential. ESCs transduced with Usp26-GFP (without Dox in medium) were collected as control (Day 0), other samples were collected at day 2, day 4 and day 6 with Dox in ES medium. (Page 18

12. *Comment:* About Fig 4b, the authors stated 'Only undifferentiated ESCs expressed CBX7 without expression of USP26, suggesting that the interaction between USP26 and CBX7 observed in 293T cells is non-specific.' This does not need to be called 'non-specific'. If there are cells which express endogenous Cbx7 and USP26, they might interact.

Response: The statement is not accurate, as suggested, so we deleted it in the main text.

13. *Comment:* Related Fig 5b, what the authors need to show is if Usp26 KD decreases CBX4 and CBX6.

Response: We performed western blotting (as reviewer suggested) to analyze protein levels in Usp26 KD MEF cells. The result showed that Cbx4 and Cbx6 protein levels decreased after Usp26 knockdown in MEF cells (Supplementary Figure S5a and S5b). See Page 9 Line 1-4.

Figure S5a and S5b. MEFs were transduced with mouse Usp26 shRNA or control shRNA lentivirus, cell extracts were harvested, and mRNA levels were analyzed by qPCR (a). Protein levels were analyzed by western blotting with specific antibodies (b).

14. *Comment:* Fig 5c, CBX6 up-regulation seems to take a long time, and does not look like a direct consequence of exogenous Usp26 expression. During RA-induced ESC differentiation where Usp26 expressing goes up, do CBX4 and 6 expression level increase? If so, this could be a result of

USP26-induced differentiation.

Response: Our result showed Cbx4 and Cbx6 protein levels increased with Usp26 during RA-induced differentiation. This could be caused by differentiation. Usp26 expression also stabilized Cbx4 and Cbx6 to further increase their protein amounts. We also propose that there is a feedback loop during differentiation, Usp26, Cbx4 and Cbx6 expression increased. The stabilized Cbx4 and Cbx6 occupied Cbx7 promoter to suppress Cbx7 expression. Decreased Cbx7 expression level caused less repression of Cbx4 and Cbx6 promoters, further enhanced Cbx4 and Cbx6 expression (Supplementary Figure S5d). See Page 9, Lines 4-6

Figure S5d. Mouse ES cells were infected with or without RA treatment, and cell extracts were harvested at different time points, protein levels were analyzed by western blotting using specific antibodies, as indicated.

15. *Comment:* Supplementary Fig. S5b. Proteasome inhibitors do not seem to increase CBX4 and CBX6 in 293T cells in the absence of exogenous USP26. It suggests that ubiquitination-mediated degradation is not active in this cell line. Why the proteasome inhibitors' effects become obvious only when exogenous USP26 is expressed? I expect that exogenous USP26 remove ubiquitin, stabilize CBX4, CBX6, therefore the proteasome inhibitors have less impact on the CBX4, CBX6

protein level. Anyway, this experiment needs to be done in MEFs.

Response: We repeated this experiment MEF cells, which express Usp26 endogenously, loading amount also reduced to avoid saturation. CBX4 and CBX6 protein levels were slightly increased upon addition of the proteasome inhibitors, MG132 or Lactacystin in MEF cells. However, no additional increase for Cbx4 or Cbx6 was observed in USP26 overexpression MEF cells, indicating endogenously expressed Usp26 is sufficient to stabilize Cbx4 and Cbx6 in MEF cells

(Supplementary Figure S5f). See Page 9 Lines 8-11.

MEF cells

Figure S5f. MEF cells with or without Usp26 lentivirus transduction were treated with DMSO, MG132, Lactacystin, or PR619. Specific proteins were analyzed by western blotting using anti-CBX4, anti-CBX6 and anti-USP26 antibodies. β -ACTIN was used as a loading control.

16. *Comment:* Supplementary Fig S6. It is important to show protein levels of CBX4, 6 in RA differentiation, since USP26 controls CBX4, 6 levels at the post translational level.

Response: We showed this protein level analysis in Supplementary Fig. S5d. Cbx4 and Cbx6 protein levels increased during RA-induced ESC differentiation. Usp26 expression stabilized Cbx4

and Cbx6 to further increase their protein amounts. (Supplementary Figure S5d). Page 9, Lines 4-7

d mESCs

Figure S5d. Mouse ES cells were infected with or without RA treatment, and cell extracts were harvested at different time points, protein levels were analyzed by western blotting using specific antibodies.

17. *Comment:* Authors stated 'Increased Cbx4 and Cbx6 mRNA was also observed in Usp26-induced or RA-induced ESC differentiation (Supplementary Fig. S6b, c)' S6c seems to be a wrong figure. Supplementary Fig. S7b. This is also not clear how day 0 colonies are maintained in the presence of the transgenes.

Response: We deleted Supplementary Fig. S6b and c to avoid misunderstanding. Page 44

For Cbx4 and Cbx6 overexpression, we have addressed in previous response of *Comment 2*, The C terminal GFP-tagged mCbx4 or mCbx6 were also cloned as same method as Usp26. We have added these details in Methods section. (Page 18 line 21- Page 19 line 2).

Figure S6b&c. (b) qPCR analysis of mouse Cbx4 and Cbx6 mRNA expression in ESCs transduced with mouse Usp26 or empty vector (EV) (** $p < 0.01$ compared to EV at day 6). (c) qPCR analysis of mouse Cbx4 and Cbx6 mRNA expression in ESCs induced with RA (RA+) or without RA (RA-) (** $p < 0.01$ compared to RA- treatment at day 6).

18. *Comment:* Discussion 'Usp26 overexpression in OSKM-induced reprogrammed MEFs did not significantly increase Cbx4 or Cbx6 gene expression, but increased CBX4 and CBX6 protein expression, suggesting that USP26-mediated deubiquitination of CBX4 and CBX6 is the predominant mechanism for CBX4 and CBX6 accumulation during somatic cell reprogramming.' I am not sure which data showed 'increased CBX4 and CBX6 protein expression' during reprogramming.

Response: We have removed this statement from the main text.

References

- 1 Li, Z., Wang, D., Messing, E. M. & Wu, G. VHL protein-interacting deubiquitinating enzyme 2 deubiquitinates and stabilizes HIF-1alpha. *EMBO Rep* **6**, 373-378, doi:10.1038/sj.embor.7400377 (2005).
- 2 Mathieu, J. *et al.* Hypoxia-inducible factors have distinct and stage-specific roles during

- reprogramming of human cells to pluripotency. *Cell Stem Cell* **14**, 592-605, doi:10.1016/j.stem.2014.02.012 (2014).
- 3 Cunningham, T. J. & Duester, G. Mechanisms of retinoic acid signalling and its roles in organ and limb development. *Nat Rev Mol Cell Biol* **16**, 110-123, doi:10.1038/nrm3932 (2015).
 - 4 Takahashi, K. *et al.* Induction of pluripotent stem cells from adult human fibroblasts by defined factors. *Cell* **131**, 861-872, doi:10.1016/j.cell.2007.11.019 (2007).

Reviewers' comments:

Reviewer #1 (Remarks to the Author):

The authors have answered the main body of reviewers' comments/questions and the manuscript is significantly improved, conveying the main findings in a clear way.

A couple of points:

Pg11. Supplementary Fig. 6d: does USP26 protein bind its own gene? In that case the observed decrease in H2Aub might be because of USP26 binding and enzymatic activity. The authors should comment on this.

Pg. 11, lines 9-15, Figure 6a and response to the review: Note sure the authors' comment that USP26 is "particularly enriched in CBX7 locus compared to SOX2 and Nanog" is correct especially if one compares days 0 and 6 or RA-induced differentiation.

Pg11, lines 1-5 and Figure S6e-g: These panels miss the control shRNA condition that should be added here.

Reviewer #2 (Remarks to the Author):

The author made some improvement on the manuscript, and the biochemical analyses have been done well. But I do not think my concern was fully addressed.

The reprogramming method – 'and then reseeded on irradiated feeder cells at the desired density' What is the desired density? No none can reproduce this experiment with this explanation.

Overexpression experiments with pTet-O-mUsp26-GFP, pTet-O-mCbx4-GFP, pTet-O-mCbx6-GFP (Fig 2f) – It is GFP tagged. Why day 0 without dox has GFP expression?

Circos plot – I still cannot understand what the % and scales mean.

Figure S5f – Why PR-619 has no effect? I do not understand the logic behind the conclusion from this data 'In MEF cells, which express USP26 endogenously, we found that proteasome inhibitors have less effect on CBX4 and CBX6 accumulation (Supplementary Fig. S5f), indicating that endogenous USP26 is sufficient to stabilize CBX4 and CBX6 by reducing proteasomal degradations'. Why endogenous USP26 expression makes the proteasome inhibitors less effective? What is the USP26 expression levels in HEK?

Fig. S7b – It is not clear how the experiments have been done. Which culture condition is it? It is not convincing without quantification. It also applies to Figure 2f, 2g, S2a. One can take images of colonies which fit with the authors' claim.

Reviewer #1

1. Pg11. Supplementary Fig. 6d: does USP26 protein bind its own gene? In that case the observed decrease in H2Aub might be because of USP26 binding and enzymatic activity. The authors should comment on this.

Response: Thank you for this thoughtful comment. We analyzed USP26 protein binding to the *Usp26* promoter in undifferentiated and RA-induced differentiated ES cells, and found no appreciable difference in *Usp26* binding between treated and untreated cells (revised Supplementary Figure S6d). Although *Usp26* protein level is high in differentiated ES cells, neither *Usp26* nor its partners CBX4 and CBX6 could bind to the *Usp26* promoter, indicating that USP26 expression is regulated by other different mechanisms.

Figure S6d: ChIP-PCR analysis of the *Usp26* promoter region in during ESC differentiation upon LIF withdrawal and RA treatment (** $p < 0.01$ compared to day 0). Red arrows indicate ChIP-PCR primer targets. Black arrow indicates the *Usp26* promoter transcriptional start site (TSSs).

2. Pg. 11, lines 9-15, Figure 6a and response to the review: Note sure the authors' comment that USP26 is "particularly enriched in CBX7 locus compared to SOX2 and Nanog" is correct especially if one compares days 0 and 6 or RA-induced differentiation.

Response: We thank the reviewer for the comment. The previous statement was not accurate, and therefore has been deleted from the revised manuscript.

3. *Pg11*, lines 1-5 and Figure S6e-g: These panels miss the control shRNA condition that should be added here.

Response: We have added control shRNA results to revised Supplementary Figure S6e & S6f. In Supplementary Figure S6g, *Usp26*, *Usp26* and *Cbx4*, or *Usp26* and *Cbx6* expression vectors or an empty vector control (EV) were transfected into cells.

Figure S6e & S6f: ChIP-PCR analysis of *Sox2*, *Nanog* and *Cbx7* promoters in ECs transduced with control shRNA, *Cbx4* shRNA *Cbx6* shRNA or *Cbx4* and *Cbx6* shRNA before and after LIF withdrawal and RA treatment (* $p < 0.5$, ** $p < 0.01$ compared to day 0).

Figure S6g: Luciferase activities of Oct4, Sox2, Nanog, and Cbx7 promoters with empty vector (EV), Usp26, or Usp26 and Cbx4 or Cbx6 co-transfection (** p<0.01 compared to EV).

Reviewer #2

1. *The reprogramming method – ‘and then reseeded on irradiated feeder cells at the desired density’ What is the desired density? No none can reproduce this experiment with this explanation.*

Response: For the reprogramming efficiency experiment, 3×10^5 feeder cells (irradiated MEFs) seeded into each well of a 6-well plate previously coated with 0.1% gelatin. We have revised this statement in the **Methods section**.

2. *Overexpression experiments with pTet-O-mUsp26-GFP, pTet-O-mCbx4-GFP, pTet-O-mCbx6-GFP (Fig 2f) – It is GFP tagged. Why day 0 without dox has GFP expression?*

Response: We found that after GFP-Usp26 transduction and Dox induction, GFP-negative (i.e. untransfected) ESCs could not be differentiated (**Figure 1, lower left colony below**). For this reason, we pre-selected GFP-positive colonies using low concentration Dox (0.1 $\mu\text{g/ml}$), and then used a higher Dox concentration (2 $\mu\text{g/ml}$) to induce ESC differentiation.

Figure 1. ESC colonies transduced with Usp26-GFP after 4 days of Dox induction. GFP-positive colonies (upper right) differentiated after Dox induction, while GFP-negative colonies (lower left) retain ESC morphology.

3. *Circos plot – I still cannot understand what the % and scales mean.*

Response: In the circos plot diagram (Figure S3), the percentages and scales represent the relative expression ranks of pluripotency genes. The percentage and scales are calculated by following formula, (percent of gene X at day N) = [expression of gene X at day N] / [the sum of all gene expression at day N]¹.

4. *Figure S5f – Why PR-619 has no effect? I do not understand the logic behind the conclusion from this data ‘In MEF cells, which express USP26 endogenously, we found that proteasome inhibitors have less effect on CBX4 and CBX6 accumulation (Supplementary Fig. S5f), indicating that endogenous USP26 is sufficient to stabilize CBX4 and CBX6 by reducing proteasomal degradations’. Why endogenous USP26 expression makes the proteasome inhibitors less effective? What is the USP26 expression levels in HEK?*

Response: Thank the reviewer for the thoughtful comments. After literature review on PR-619, we found that PR-619 effects on protein degradation are complicated and dependent on cell types. We, therefore, removed the data related to PR-619 treatment, and revised the text accordingly on **Page 9, line 10-13**. Furthermore, we added a western blot of Usp26 expression level in HEK293T cells in the **updated Figure S5e**.

e 293T cells

Figure S5e. 293T cells were treated with DMSO, MG132 or Lactacystin. Specific proteins were analyzed by western blotting using anti-CBX4 and anti-CBX6 antibodies. β-actin was used as a loading control.

5. *Fig. S7b – It is not clear how the experiments have been done. Which culture condition is it? It is not convincing without quantification. It also applies to Figure 2f, 2g, S2a. One can take images of colonies which fit with the authors' claim.*

Response: More detailed experiment procedures for the culture conditions are now included in the **Methods section**:

Reprogramming and shRNA knockdown (**Figure 1b, S7b**): Tet-O inducible Oct4-Sox2-Klf4-cMyc (OSKM) MEFs were used to generate iPSCs. OSKM transgenic MEF cells were transduced with GIPZ lentivirus-based shRNAs targeting USP family members, Cbx4, Cbx6, or Cbx4 and Cbx6. These cells were selected with 2 µg/ml puromycin for 2 days. Before reprogramming, 3×10^5 feeder cells (irradiated MEFs) per well were seeded into 6-well plates previously coated with 0.1% gelatin. 1000 puromycin selected OSKM transgenic MEF cells were reseeded onto feeder cells in 6-well plates. The next day, modified iSF1 medium (DMEM supplemented with 10% Knockout serum, 2 mM L-glutamine, 100 µM non-essential amino acids (Gibco), 0.1 mM β-mercaptoethanol, 50 ng/ml LIF, 5 µg/ml CHIR99021, 2.5 µg/ml PD0325901) containing 2 µg/ml Dox was added and replenished every day. The efficiency of iPSC formation was calculated based on the number of AP⁺ iPSC colonies. The colonies were stained for AP activity on day 12 with AP detection Red Substrate kit (Vector). To fully characterize iPSCs, these cells were fixed with methanol for 20 min at -20 °C, and nonspecific receptors were blocked with 10% normal goat serum. Cells were then hybridized with anti-Oct4, anti-Nanog and anti-Dppa4 specific antibodies (Santa Cruz Biotechnology) and then hybridized with a goat anti-mouse antibody conjugated with Texas Red (Invitrogen) and nuclei were stained with 4,6-diamidino-2-phenylindole (DAPI; Invitrogen). AP⁺ colonies were counted with a bright field microscope and Nanog⁺ colonies were counted with an Olympus 1X71S1F fluorescence microscope.

ES cells differentiation (**Figure 2f, 2g, S2a**): For Usp26 induced differentiation, ES cells were transduced with Dox-inducible GFP-tagged Usp26 or empty vector lentivirus. After 1 day of selection with 0.1 µg/ml Dox, GFP-positive colonies were cultured in iSF1 medium (2 µg/ml Dox) as described above, and individual colonies were tracked and photographed over 4 days. For RA induced differentiation, Usp26 KO, Cbx4 KO, Cbx6 KO or WT ES cells were cultured

in ES differentiation medium (DMEM supplemented with 10% Knockout serum, 2 mM L-glutamine, 100 μ M non-essential amino acids (Gibco), 0.1 mM β -mercaptoethanol) with 1 μ M RA. Individual colonies were tracked and photographed over 8 days.

References:

1. Morabito, F. *et al.* Clinical monoclonal B lymphocytosis versus Rai 0 chronic lymphocytic leukemia: A comparison of cellular, cytogenetic, molecular, and clinical features. *Clin Cancer Res* **19**, 5890–5900 (2013).
2. Altun, M. *et al.* Activity-Based Chemical Proteomics Accelerates Inhibitor Development for Deubiquitylating Enzymes. *Chem. Biol.* **18**, 1401–1412 (2011).
3. Lee, J.-G., Baek, K., Soetandyo, N. & Ye, Y. Reversible inactivation of deubiquitinases by reactive oxygen species in vitro and in cells. *Nat Commun* **4**, 1568 (2013).
4. Seiberlich, V., Borchert, J., Zhukareva, V. & Richter-Landsberg, C. Inhibition of Protein Deubiquitination by PR-619 Activates the Autophagic Pathway in OLN-t40 Oligodendroglial Cells. *Cell Biochem Biophys* **67**, 149–160 (2013).
5. Seiberlich, V., Goldbaum, O., Zhukareva, V. & Richter-Landsberg, C. The small molecule inhibitor PR-619 of deubiquitinating enzymes affects the microtubule network and causes protein aggregate formation in neural cells: Implications for neurodegenerative diseases. *Biochimica et Biophysica Acta (BBA) - Molecular Cell Research* **1823**, 2057–2068 (2012).

Reviewers' comments:

Reviewer #1 (Remarks to the Author):

The authors have sufficiently answered the reviewers' questions and, I think, the manuscript has the scientific merit for publication in Nature Communications.

Reviewer #2 (Remarks to the Author):

Response: We found that after GFP-Usp26 transduction and Dox induction, GFP-negative (i.e. untransfected) ESCs could not be differentiated (Figure 1, lower left colony below). For this reason, we pre-selected GFP-positive colonies using low concentration Dox (0.1 µg/ml), and then used a higher Dox concentration (2 µg/ml) to induce ESC differentiation.

This explanation seem to make more sense than previous explanation, but I doubt it was how this experiment had been done. First, now the first picture of the time course became day 1, instead of day 0 in the previous manuscript. If the first image is day 1, there is no need to expose the cells with low Dox. Pictures of the green colonies could be taken on day 1 without 'pre-selection'. The size of day 1 colonies also does not look like that of ESCs cultured for only 1 or 2 days. My guess is that infected ESCs were cultured for several day to let them form colonies, then dox was added (day0). Day 1 is first time point of imaging. I might be wrong, but the explanation, images and the date labelling do not make sense.

Circos plot – I still do not understand. How did the author combine the data from ESC differentiation and reprogramming experiments? Why don't the authors provide the expression values of all tested 90 genes in all the tested conditions? It is unlikely that expression of Nanog, Sox2, Cbx7 was affected, but not other pluripotency genes such as Esrrb, Klf2, Dppa2, Rex1 when reprogramming efficiency was enhanced. Otherwise, cells on day 12 with shUsp26 would not be iPSCs. In addition, Sox2, Nanog, Cbx7 expression looks same on day 4 in the +shUsp26 OSKM-MEF reprogramming (Fig 3a). It also let me wonder why the Circos plot shows the bold connection on day4.

Fig. S7c. It does not look CBX4, CBX6 expression induced differentiation, as Usp26 did. The colony is very similar to that of EV.

According to the methods, 1000 OSKM MEFs generated 100 iPSC colonies in the control condition. It means 10% of OSKM MEFs generated iPSC colonies (not only AP+, but also Nanog-GFP+). With shUsp26, shCbx4, shCbx6, it becomes 20-30%. It is extremely high. It is necessary to describe where the OSKM MEFs came from.

Figure S2c,d. Why Usp26 is absent in Western blotting (S2d) if the KO efficiency was so low with SURVEYIR assay (S2c)?

Response to Reviewer #2

1. *Comment: This explanation seems to make more sense than previous explanation, but I doubt it was how this experiment had been done. First, now the first picture of the time course became day 1, instead of day 0 in the previous manuscript. If the first image is day 1, there is no need to expose the cells with low Dox. Pictures of the green colonies could be taken on day 1 without 'pre-selection'. The size of day 1 colonies also does not look like that of ESCs cultured for only 1 or 2 days. My guess is that infected ESCs were cultured for several days to let them form colonies, then dox was added (day0). Day 1 is first time point of imaging. I might be wrong, but the explanation, images and the date labelling do not make sense.*

Response: Thanks for these thoughtful comments. To explain how we perform the experiment of Usp26 induced differentiation, we have added a scheme to describe experiment details (Fig. 2f). In brief, ESCs were transduced with lentiviruses expressing Dox-inducible GFP-tagged Usp26 or empty vector (Day -3). After ESCs formed colonies, 0.1 $\mu\text{g/ml}$ Dox was added for pre-selection (Day -1). Day 0 was defined as the day when pre-selected GFP positive colonies were cultured in iSF1 medium with high dose Dox (2 $\mu\text{g/ml}$), individual colonies were tracked and imaged on days 1, 2, 3 and 4. We have revised this part in the Methods section and legend of Fig. 2f & g.

Figure 2f. Experimental scheme of Usp26 induced ESC differentiation. ESCs were transduced with Dox inducible GFP-tagged Usp26 or empty vector lentivirus (Day -3). After ESCs colonies formed (Day -1), then low dose Dox (0.1 $\mu\text{g/ml}$) was added for pre-selection. Day 0 was defined as the day when pre-selected GFP positive colonies were cultured in iSF1 medium with high dose Dox (2 $\mu\text{g/ml}$), individual colonies were tracked and taken pictures on days 1, 2, 3 and 4.

2. *Comment: Circos plot – I still do not understand. How did the author combine the data from ESC differentiation and reprogramming experiments? Why don't the authors provide the expression values of all tested 90 genes in all the tested conditions? It is unlikely that expression of Nanog, Sox2, Cbx7 was affected, but not other pluripotency genes such as Esrrb, Klf2, Dppa2, Rex1 when reprogramming efficiency was enhanced. Otherwise, cells on day 12 with shUsp26 would not be iPSCs. In addition, Sox2, Nanog, Cbx7 expression looks same on day 4 in the +shUsp26 OSKM-MEF reprogramming (Fig 3a). It also let me wonder why the Circos plot shows the bold connection on day 4.*

Response: We did not combine data from ESC differentiation and reprogramming experiments. All data presented in Circos plot are from reprogramming experiments. We have now provided all gene expression data from reprogramming experiments in Supplementary Table S2. To avoid the confusion with Circos plot, we have deleted Circos plot (Supplemental Figure S3), and

included Supplemental Table S2. We found that *Sox2*, *Nanog* and *Cbx7* are the most significantly increased genes after *Usp26* knockdown at the reprogramming day 8 and day 12 (Table S2). Pluripotency genes such as *Esrrb*, *Klf2*, *Dppa2*, *Rex1* are also affected during reprogramming, but the increased levels were not as high as *Sox2*, *Nanog* and *Cbx7*, compare to those in the control shRNA group (Table. S2).

Table S2. Relative expression of the 90 marker genes during reprogramming of *Usp26* shRNA and control shRNA lentivirus-transduced OSKM-MEFs, Table S2 related to main Figure 3.

Genes	MEFs	Usp26 shRNA			Control shRNA		
		Reprogramming (Dox + days)			Reprogramming (Dox + days)		
		Day 4	Day 8	Day 12	Day 4	Day 8	Day 12
Sox2	0.1040	9.1580	312.7620	1914.3427	3.0777	12.7276	33.4164
Nanog	0.1573	5.5092	58.1025	322.6628	2.1340	13.7125	31.1184
Cbx7	0.2124	5.6848	35.8774	59.9464	2.3229	11.2119	8.1445
Nr5a2	0.3527	2.2198	66.2610	78.5334	2.4718	13.5231	31.8145
Eras	0.3755	5.8887	6.7144	61.8228	2.9274	2.8980	15.3822
Esgl1	0.4042	8.3272	11.4276	59.4766	5.0317	9.5861	15.3494
Fgf4	0.4123	14.7526	37.5496	45.1870	4.3677	14.4847	21.4441
Cdh1	0.4223	7.1680	10.7094	45.0102	3.1852	5.4547	26.1039
Oct4	0.4316	5.2458	14.8684	34.2242	2.4463	8.8868	33.4922
Tcf1	0.4436	1.3007	10.2517	32.5607	1.2636	10.7496	36.6170
Cripto	0.5044	4.5962	16.5084	29.9862	3.7025	14.5916	23.3144
Rex1	0.5494	5.0549	14.7512	29.5995	2.3229	5.2117	13.5188
c-Myc	0.5581	3.6610	13.0054	28.2707	2.3169	7.5098	14.3036
Tet2	0.5757	1.2945	1.1254	23.5568	2.2225	1.1387	12.8672
Cldn4	0.5840	2.3098	13.1026	20.5315	1.9551	11.1266	20.4850
Cldn11	0.6043	9.9459	8.1975	20.0952	9.4711	10.9216	20.0120
Dppa2	0.6137	2.6025	3.3340	18.9144	1.4446	2.5387	9.0640
Lin28	0.6427	8.3690	19.4537	17.9600	5.5735	21.8878	12.0691
Klf4	0.6519	7.5632	5.4732	17.2102	9.9906	4.4546	8.6193
Tcf1	0.6606	3.3237	12.4449	17.1873	1.8458	10.1099	12.4109
Sall4	0.6625	4.2511	6.5654	14.9814	2.4029	6.5035	12.6369
Crb3	0.6756	3.4340	1.8614	13.9856	6.7436	2.0445	13.2472
Cldn3	0.6762	2.5047	8.4801	13.2494	2.0546	3.0620	7.8685
Klf2	0.6919	7.6411	11.1017	13.2358	2.0406	5.0289	4.8216
Utf1	0.6951	1.7284	1.5623	12.3969	1.1174	1.5929	13.1391
Gdf3	0.6990	2.5865	9.0324	10.9406	3.2523	7.2624	9.5907
Esrrb	0.7006	2.1919	6.9002	9.5450	2.6087	2.9513	6.4905
Fbxl10	0.7209	5.7137	1.9740	7.9334	1.7086	2.0356	6.7664
Dnmt3a	0.7758	1.2327	4.2657	7.4667	3.1715	1.9840	7.6537
Epcam	0.7836	2.9126	4.6701	7.2336	1.7564	5.3614	2.3707
Gata2	0.8137	1.4613	4.2260	5.5686	1.3127	1.3293	5.9865
Tet1	0.8170	1.4418	2.5616	5.4741	2.4453	1.0310	5.6047
Gata3	0.8693	1.4497	5.0122	4.3329	2.2114	5.1498	4.1657
Tbx3	0.8717	1.7930	3.8759	3.1745	1.7198	3.3249	3.0836
Phf20	0.9614	1.7851	1.2200	3.7187	1.7897	1.8959	3.7013
Lif3	0.8724	4.4632	1.4787	2.1756	3.9425	0.9787	1.4546
Slc2a1	0.8785	0.3488	1.7512	1.9651	0.3589	1.4531	1.2396
Bub1	0.8914	0.8401	1.9386	1.9305	1.4338	3.8063	7.8389
Gata4	0.9247	1.2405	1.8087	1.7481	1.2516	0.7805	1.4801
Ccnf	0.9577	1.0277	2.0974	1.7424	1.7666	3.4963	10.9990
Grb2	0.9894	1.1641	1.7639	1.6458	1.4212	1.4850	1.7539

Ezh2	1.0008	1.0498	1.1505	1.4864	2.3407	1.7339	1.3476
Nr6a1	1.0045	2.1212	1.4276	1.3270	1.7468	1.5282	4.9936
Otnbl1	1.0262	1.6607	0.9813	1.1648	0.7564	1.0139	1.1891
Ink4a	1.0553	1.7489	0.8314	1.1290	1.5842	0.2915	0.3413
p21	1.0641	1.9681	1.3369	1.1121	2.6702	0.5855	0.3708
Cdc20	1.0736	0.7325	1.3728	1.0964	1.3123	2.2280	3.5789
Csnk2a1	1.0933	1.5969	1.0245	1.0451	1.8977	0.8055	1.3570
Gata5	1.1800	1.4559	0.9625	1.0077	1.2867	0.7231	0.4230
Gata6	1.2609	0.8577	1.1701	1.0014	1.0492	1.9048	2.3779
Notch1	1.3383	0.8557	0.7758	0.9819	0.7005	0.6193	1.6280
Rbpj	1.3727	0.5381	0.9608	0.9349	0.1932	0.1973	1.4686
Nes	1.3838	4.3288	2.0190	0.9288	5.4014	3.3360	0.4585
Tet3	1.3864	1.8770	0.6824	0.8748	1.9516	0.9112	0.5386
Kdm1	1.4693	0.9325	0.7449	0.8216	2.1902	2.3160	4.4221
Tgfb2	1.4846	2.1776	2.2124	0.8014	1.6744	1.1898	0.8634
Gsk3b	1.5757	1.3621	0.7617	0.7919	1.6584	0.5600	0.5202
Bmprla	1.9654	1.2315	0.6908	0.7909	1.0314	0.5242	0.3864
B-Catenin	2.7032	1.5043	0.9487	0.7870	1.0719	0.5175	0.5620
Arf	2.7304	0.8550	0.4287	0.7078	1.7507	0.6740	1.4102
p53	3.1819	1.1592	0.9183	0.7030	1.0531	0.6652	1.2494
Zeb1	3.2415	1.2281	0.6872	0.6591	1.6224	0.3772	0.3473
Mad21l	3.3585	1.0505	0.7048	0.6479	1.7663	0.7036	0.4926
Tgfb1	3.5221	1.3062	0.6082	0.5716	1.1682	0.4587	0.3187
Hes	3.7422	0.7390	0.4875	0.5443	0.8455	0.4271	0.2786
Dnmt3b	4.0532	1.0527	0.5038	0.5422	1.6612	0.7122	1.4460
Tcf3	4.3135	1.4483	0.9479	0.5370	0.9281	0.5894	1.5814
Myst3	4.3606	1.1864	0.4926	0.5327	1.4821	0.6442	1.0960
Ocln	4.5336	0.7702	0.9887	0.5123	0.3221	0.4075	0.7095
Zfp532	4.6793	0.9664	0.7248	0.5115	0.7442	0.7663	0.5686
Wnt1	5.2790	0.9112	0.8750	0.4947	1.8225	1.0902	0.4951
Stat3	5.4165	0.9705	0.4139	0.4922	0.8579	0.2315	0.1391
Zeb2	5.5831	1.3548	0.4618	0.4813	1.1060	0.2637	0.1696
Bmi1	5.7844	0.9646	0.3885	0.4696	1.6082	0.5966	0.5162
Dnmt1	5.8576	1.2272	0.3735	0.4340	2.3342	0.6648	0.4234
Foxp1	6.3480	1.0411	0.6874	0.4228	0.8669	0.6591	0.2097
Jag1	7.0628	1.0574	0.3714	0.3891	0.8428	0.3285	0.1622
Snail	7.2551	0.5762	0.3027	0.3690	0.7410	0.1168	0.2123
Fut4	7.3104	1.4365	0.7605	0.3592	1.1458	0.2381	0.2608
Tgfb2	7.6713	1.5543	0.4363	0.3104	1.6772	0.1142	0.1002
Slug	7.9573	0.5371	0.2133	0.2492	0.3867	0.1320	0.0799
Tgfb1	9.2710	0.6387	0.3567	0.2377	0.5874	0.1121	0.0882
Cldn7	9.2922	0.5274	0.4985	0.2291	0.7028	0.3651	1.9899
N-Cadherin	10.9125	0.7131	0.3179	0.2199	0.6321	0.2027	0.0749
Fgf5	11.1623	0.1106	0.1075	0.2126	0.2951	0.1230	0.2457
Tgfb3	12.8392	0.4139	0.2639	0.1874	0.5107	0.0757	0.0660
Tgfb3	12.9738	0.3212	0.2065	0.1684	0.4937	0.0665	0.1338
Cbx4	21.6564	0.1615	0.1486	0.1416	0.9112	0.7078	0.4287
Cbx6	23.3443	0.2951	0.1024	0.0847	0.7390	0.5162	0.2340
Usp26	53.1493	0.0835	0.1032	0.0731	0.8658	0.2569	0.1024

3. Comment: Fig. S7c. It does not look CBX4, CBX6 expression induced differentiation, as Usp26 did. The colony is very similar to that of EV.

Response: ESCs differentiation induced by CBX4 overexpression alone, or CBX6 overexpression alone appears not as strong as USP26 did, suggesting that CBX4 and CBX6 work in concert to promote ESC differentiation. This notion is further supported by the fact that knockdown of both CBX4 and CBX6 enhanced reprogramming efficiency more than either alone, and reached a level similar to that of USP26 knockdown (Fig. S6b). This may explain why either CBX4 or CBX6 overexpression alone does not promote ESC differentiation as efficiently as USP26.

4. *Comment: According to the methods, 1000 OSKM MEFs generated 100 iPSC colonies in the control condition. It means 10% of OSKM MEFs generated iPSC colonies (not only AP+, but also Nanog-GFP+). With shUsp26, shCbx4, shCbx6, it becomes 20-30%. It is extremely high. It is necessary to describe where the OSKM MEFs came from.*

Response: although we initially seeded 1000 MEFs, they grew and divided during the early stage of reprogramming. Thus, the reprogramming efficiency generally lower than 10% in control condition. The OSKM-expressing MEFs (Tet-O-OSKM MEFs) used in this study were generated by our lab, as previously described¹. These cells could be efficiently reprogrammed to generate iPSCs in the presence of Dox.

5. *Comment: Figure S2c,d. Why Usp26 is absent in Western blotting (S2d) if the KO efficiency was so low with SURVEYOR assay (S2c)?*

Response: SURVEYOR assay is a semi-quantitatively method to detect indels. The visual band intensity of SURVEYOR assay is not directly correlated with knockout efficiency of guide RNA². We tested 2 individual sgRNAs by SURVEYOR assay, and sgRNA1 appeared to be more efficient at producing indels than sgRNA2 (Figure S2c). To determine knockout efficiency, we used western blotting analysis, and found that sgRNA1 could knockout Usp26 efficiently (Figure S2d).

Reference:

1. Zhao, W. *et al.* Jmjd3 inhibits reprogramming by upregulating expression of INK4a/Arf and targeting PHF20 for ubiquitination. **152**, 1037–1050 (2013).
2. Qiu, P. *et al.* Mutation detection using Surveyor nuclease. *BioTechniques* **36**, 702–707 (2004).

REVIEWERS' COMMENTS:

Reviewer #2 (Remarks to the Author):

Regarding my comment 1, Fig 2g is now explained better by Fig 2f, but it is what I guessed. They have claimed something different until the previous revision. I felt ones who wrote the manuscript did not fully understand how the experiments were done. After 3 rounds of revision, I still feel the same. For example, 'Furthermore, Cbx4 or Cbx6 alone could inhibit increased reprogramming efficiency mediated by Usp26 shRNA, thus suggesting that Usp26 regulates Cbx4 or Cbx6, which, in turn, downregulates expression of pluripotency genes (Supplementary Fig. S6b)'. I cannot find such experiments.....

Regarding my comment 2, I asked how data from ESC differentiation and reprogramming were combined based on the sentence 'To study the effects of Usp26 knockdown on gene expression during MEF reprogramming and ESC differentiation, we performed qPCR and measured the expression levels of 90 genes, which have been reported to be involved in reprogramming and ESC pluripotency (Supplementary Fig. S3).' I agree that it is better not to use the Circos plot which it was not explained well, but the sentence is still 'To study the effects of Usp26 knockdown on gene expression during MEF reprogramming and ESC differentiation, we performed qPCR and measured the expression levels of 90 genes, which have been reported to be involved in reprogramming and ESC pluripotency (Supplementary Table. S2).' The Supplementary table 2 does not have any data from ESC differentiation.

Regarding my comment 3, authors admitted that differentiation induction by Cbx4 or Cbx6 were not as strong as that by Usp26. However, the text is still 'We also showed that Cbx4 and Cbx6 could promote ESC differentiation in Usp26 knockout cells (Supplementary Fig. S6c), similar to results obtained by overexpressing Usp26 (Fig. 2f).' (This Fig 2f is a mistake; it should be Fig 2g). I asked quantification from the 1st round of the revision, but I still don't see it. In addition, Cbx4/6 overexpression was in Usp26 KO ESCs, Usp26 overexpression in Fig 2g was in WT ESCs. I thought what they meant was Figure 3c. However the text says 'Overexpression of Usp26 in ESCs led to decreased mRNA levels of Sox2, Nanog, and Oct4 compared with control cells (Fig. 3c)', while the figure says (RA+day) on the X axis. I cannot know what condition was used in which experiments.

Regarding my comment 4. Of course cells divide during reprogramming. However, if one seeds only 1000 OSKM MEFs in a 6 well plate, it is likely that most of colonies are from a single MEFs. The protocol does not contain any harvest/re-seeding process. From the beginning I wondered if all of the ALP+ colonies were real iPSC colonies, and asked to use specific markers and perform quantification. They included pluripotency markers, but did not perform any quantification. I am still wonder if ALP colonies counting reflects iPSC colony numbers.

Regarding my comment 5. I know SURVEYOR assay is semi-quantitative, but the data is too far from the Western blotting. They should have used IDAA (<https://doi.org/10.1093/nar/gkv126>) or TIDE ([doi: 10.1093/nar/gku936](https://doi.org/10.1093/nar/gku936)) assay for indel quantification.

Thank you for your helpful comments and suggestions on our manuscript "**Usp26 Functions as a Negative Regulator of Cellular Reprogramming by Stabilising PRC1 Complex Components**". We have addressed the comments from Reviewer # 2 by changing the wording, providing the requested quantifications, removing the SURVEYOR assay data, and using TIDE assay suggested by the reviewer. All suggested changes from reviewer #2 in the revised manuscript and supplemental materials have been marked in yellow.

REVIEWERS' COMMENTS:

Reviewer #2 (Remarks to the Author):

Comments: Regarding my comment 1, Fig 2g is now explained better by Fig 2f, but it is what I guessed. They have claimed something different until the previous revision. I felt ones who wrote the manuscript did not fully understand how the experiments were done. After 3 rounds of revision, I still feel the same. For example, 'Furthermore, Cbx4 or Cbx6 alone could inhibit increased reprogramming efficiency mediated by Usp26 shRNA, thus suggesting that Usp26 regulates Cbx4 or Cbx6, which, in turn, downregulates expression of pluripotency genes (Supplementary Fig. S6b)'. I cannot find such experiments.....

Response: We have modified the sentence to "Furthermore, Cbx4 or Cbx6 alone could inhibit increased reprogramming efficiency mediated by Usp26 shRNA (Supplementary Fig. S6b)." in main text **Page 12, Line 7-8.**

Comments: Regarding my comment 2, I asked how data from ESC differentiation and reprogramming were combined based on the sentence ' To study the effects of Usp26 knockdown on gene expression during MEF reprogramming and ESC differentiation, we performed qPCR and measured the expression levels of 90 genes, which have been reported to be involved in reprogramming and ESC pluripotency (Supplementary Fig. S3).' I agree that it is better not to use the Circos plot which it was not explained well, but the sentence is still ' To study the effects of Usp26 knockdown on gene expression during MEF reprogramming and ESC differentiation,

we performed qPCR and measured the expression levels of 90 genes, which have been reported to be involved in reprogramming and ESC pluripotency (Supplementary Table. S2).’ The Supplementary table 2 does not have any data from ESC differentiation.

Response: We have modified this sentence to “To study the effects of Usp26 knockdown on gene expression during MEF reprogramming, we performed qPCR and measured the expression levels of 90 genes, which have been reported to be involved in reprogramming (Supplementary Table. S2).” in **main text Page 7, Line 5-7**

Comments: Regarding my comment 3, authors admitted that differentiation induction by Cbx4 or Cbx6 were not as strong as that by Usp26. However, the text is still ‘We also showed that Cbx4 and Cbx6 could promote ESC differentiation in Usp26 knockout cells (Supplementary Fig. S6c), similar to results obtained by overexpressing Usp26 (Fig. 2f).’ (This Fig 2f is a mistake; it should be Fig 2g). I asked quantification from the 1st round of the revision, but I still don’t see it. In addition, Cbx4/6 overexpression was in Usp26 KO ESCs, Usp26 overexpression in Fig 2g was in WT ESCs. I thought what they meant was Figure 3c. However the text says ‘Overexpression of Usp26 in ESCs led to decreased mRNA levels of Sox2, Nanog, and Oct4 compared with control cells (Fig. 3c)’, while the figure says (RA+day) on the X axis. I cannot know what condition was used in which experiments.

Response: We have corrected “Fig. 2f” to “Fig. 2g” **in the main text (Page 12, Line 12)**. This ES differentiation experiment was performed with Usp26 overexpression and not RA induction. Furthermore, labels in Fig. S6c, Fig.2g, and Fig. 3c are overexpression. We didn’t mention RA induction in these three figures. We have added quantification analysis of ESCs in **Supplementary Figure S2b and S3c.**

Supplementary Figure S2b. Quantification of ESC colonies transduced with Dox inducible GFP-tagged *mUsp26* overexpression or with GFP-tagged empty vector (**** $p < 0.0001$ compared to empty vector, $n=3$).

Supplementary Figure S2c. Quantification of ESC colonies during LIF withdrawal (LIF-) and RA-induced (RA+) differentiation in wild-type (WT) and *Usp26* knockout (KO) ES cells (** $p < 0.01$, **** $p < 0.0001$ compared to WT, $n=3$).

Comments: Regarding my comment 4. Of course cells divide during reprogramming. However, if one seeds only 1000 OSKM MEFs in a 6 well plate, it is likely that most of colonies are from a single MEFs. The protocol does not contain any harvest/re-seeding process. From the beginning I wondered if all of the ALP+ colonies were real iPSC colonies, and asked to use specific markers and perform quantification. They included pluripotency markers, but did not perform any quantification. I am still wonder if ALP colonies counting reflects iPSC colony numbers.

Response: We added the MEFs isolation protocol in the **Supplementary Methods section**.

Sacrifice a pregnant OSKM-mouse at 13 or 14 day post-coitum by cervical dislocation. Dissect out the uterine horns and separate embryos then dissect head and red organs, wash in PBS and finely mince the tissue with a sterile razor blade until it becomes possible to pipette. Trypsin each embryo with 1 ml of 0.05% trypsin/EDTA (Gibco, Invitrogen) for 15 min at 37 °C. Plate cells from 3 embryos in each T175 flask for 24 hours (P0). Expand P0 cells till P2 or P3, then cells were frozen for future usage.

Actually, we did provide NANOG positive colony quantification in **Supplementary Figure S1c** of the previous revision. This result showed that reprogramming efficiency of OSKM-MEFs transduced with *Usp26* shRNA is about 3-fold compare to OSKM-MEFs transduced with control shRNA. To strengthen our results, we also provide another stem cell maker SSEA1 staining quantification analysis in **Supplementary Figure S1d**.

Supplementary Figure S1c&d. Quantification of Nanog⁺ (c) and SSEA1 (d) colonies after 12 days of OSKM induction in MEFs transduced with control or *Usp26* shRNA (p<0.01 compared to control shRNA)**

Comments: Regarding my comment 5. I know SURVEYOR assay is semi-quantitative, but the data is too far from the Western blotting. They should have used IDAA (<https://doi.org/10.1093/nar/gkv126>) or TIDE (doi: 10.1093/nar/gku936) assay for indel quantification.

Response: We have removed SURVEYOR assay and added TIDE analysis¹ of CRISPR/Cas9 knockout in *Usp26* in MEF cells in **Supplementary Figure S2d** and the **Supplementary methods section**.

Supplementary Figure S2d. TIDE assay of CRISPR/Cas9 knockout in *Usp26* in MEF cells

Reference:

1. Brinkman, E. K., Chen, T., Amendola, M. & van Steensel, B. Easy quantitative assessment of genome editing by sequence trace decomposition. *Nucleic Acids Res.* **42**, e168–e168 (2014).

Thank you for the opportunity to improve the quality of our manuscript by incorporating these suggested changes.